# Mec1-independent activation of the Rad53 checkpoint kinase revealed by quantitative analysis of protein localization dynamics

Brandon Ho[1,2], Ethan J Sanford[3], Raphael Loll-Krippleber[1,2], Nikko P Torres[1,2], Marcus B Smolka[3], Grant W Brown[1,2]*

[1]Department of Biochemistry, University of Toronto, Toronto, Canada; [2]Donnelly Centre for Cellular and Biomolecular Research, University of Toronto, Toronto, Canada; [3]The Weill Institute for Cell & Molecular Biology, Cornell University, Ithaca, United States

**\*For correspondence:**
grant.brown@utoronto.ca

**Competing interest:** The authors declare that no competing interests exist.

**Abstract** The replication checkpoint is essential for accurate DNA replication and repair, and maintenance of genomic integrity when a cell is challenged with genotoxic stress. Several studies have defined the complement of proteins that change subcellular location in the budding yeast *Saccharomyces cerevisiae* following chemically induced DNA replication stress using methyl methanesulfonate (MMS) or hydroxyurea (HU). How these protein movements are regulated remains largely unexplored. We find that the essential checkpoint kinases Mec1 and Rad53 are responsible for regulating the subcellular localization of 159 proteins during MMS-induced replication stress. Unexpectedly, Rad53 regulation of the localization of 52 proteins is independent of its known kinase activator Mec1, and in some scenarios independent of Tel1 or the mediator proteins Rad9 and Mrc1. We demonstrate that Rad53 is phosphorylated and active following MMS exposure in cells lacking Mec1 and Tel1. This noncanonical mode of Rad53 activation depends partly on the retrograde signaling transcription factor Rtg3, which also facilitates proper DNA replication dynamics. We conclude that there are biologically important modes of Rad53 protein kinase activation that respond to replication stress and operate in parallel to Mec1 and Tel1.

## Editor's evaluation

This fundamental study identifies a novel non-canonical activation mode for the central checkpoint kinase Rad53, a phosphorylation event that does not depend on Mec1 and instead depends on proteins involved in retrograde signaling through Rtg3. The evidence supporting the conclusion is compelling, with rigorous genetic, imaging, and proteomic approaches. Collectively, the findings convincingly demonstrate unanticipated complexities in the cellular DNA replication stress response, which will be of broad interest to the genome stability field.

## Introduction

The DNA replication machinery serves to faithfully duplicate the genome in a highly regulated manner. Consistent with DNA replication being an essential cellular process, failure to replicate the genome leads to several physiological defects, including development disorders, neurodegeneration, growth defects, and predisposition to cancer (*Aguilera and García-Muse, 2013*; *Ubhi and Brown, 2019*; *Zeman and Cimprich, 2014*). Endogenous and exogenous forms of DNA damage can act as

obstacles that impair the processivity of the replication fork machinery, resulting in DNA replication stress. Chemically inducing methylation of nucleotide bases with methyl methanesulfonate (MMS) or depleting the nucleotide levels within a cell using hydroxyurea (HU) have both been shown to stall the replication fork (*Pellicioli et al., 1999*; *Tercero et al., 2003*). All forms of replication stress result in stretches of single-stranded DNA (ssDNA) that is rapidly coated with replication protein A complex (RPA), the signal for replication checkpoint activation.

The eukaryotic replication checkpoint serves to promote survival in conditions of replication stress by preventing cell-cycle progression and increasing nucleotide levels to allow for proper DNA repair. In the budding yeast, *Saccharomyces cerevisiae*, RPA-coated ssDNA is recognized and bound by Ddc2, which recruits its binding partner and an essential replication checkpoint kinase, Mec1 (*Lisby et al., 2004*; *Pardo et al., 2017*). Several protein factors at stressed forks, including the Rad17/Mec3/Ddc1 complex and Dpb11, activate and/or potentiate Mec1 kinase activity. Mec1 subsequently phosphorylates histone H2A at Ser129, promoting the recruitment of a mediator protein Rad9, which is then phosphorylated (*Gilbert et al., 2001*). Mec1 can also phosphorylate the checkpoint mediator and component of the replication fork complex Mrc1, which causes dissociation from Pol ε and mediates Rad53 phosphorylation by Mec1 (*Alcasabas et al., 2001*; *Chen et al., 2014*; *Zou and Elledge, 2003*). Both mediators are capable of binding Rad53 to increase its local protein concentration, stimulating Rad53 trans-autophosphorylation and full activation (*Branzei and Foiani, 2006*; *Wybenga-Groot et al., 2014*). Hyperphosphorylated Rad53 then amplifies and transduces the signal cascade through several of its substrates, including another checkpoint kinase, Dun1 (*Chen et al., 2007*; *Lee et al., 2003*).

Numerous replication stress and DNA repair proteins organize into multiprotein complexes that can be observed as discrete nuclear foci using fluorescence microscopy (*Gallina et al., 2015*; *Lisby et al., 2003*; *Tkach et al., 2012*). This regulated re-localization of proteins is a key feature of the replication checkpoint response. Indeed, several proteins involved in Mec1 activation (Ddc1, Ddc2, and Dpb11) form subnuclear foci when cells are treated with MMS or HU (*Bonilla et al., 2008*; *Lisby et al., 2004*). In addition, the spatiotemporal organization of recombination repair proteins into foci has proven to be a valuable tool in dissecting the genetic requirements for the order of assembly of repair foci (*Gallina et al., 2015*; *Tkach et al., 2012*). Several studies have expanded on the repertoire of proteins that relocalize during replication stress by utilizing the budding yeast GFP collection to systematically probe protein subcellular location during replication stress conditions (*Breker et al., 2013*; *Chong et al., 2015*; *Dénervaud et al., 2013*; *Mazumder et al., 2013*; *Tkach et al., 2012*). We previously assessed protein location by manual inspection after treatment with HU or MMS and observed 254 proteins changing localization (*Tkach et al., 2012*). Interestingly, protein movement involved almost every cellular compartment, covering 10 localization categories, with many localizations not involving the nucleus (*Tkach et al., 2012*). Finally, we, and others, have identified two distinct classes of nuclear foci. These include the DNA damage repair foci linked to DNA repair processes, and the intranuclear quality control compartments (INQ) involved in the turnover of replication stress factors, demonstrating that protein localization can provide insight into functional processes (*Gallina et al., 2015*; *Tkach et al., 2012*). While replication stress-induced protein localization has been well documented and has provided functional insight, the details of how these movements are regulated remains elusive.

Mec1 and Rad53 have a combined total of ~1200 protein targets that they phosphorylate during replication stress and DNA damage conditions (*Bastos de Oliveira et al., 2015*; *Chen et al., 2007*; *Lanz et al., 2018*), and are therefore prime candidates for regulators of protein localization changes during replication stress. In this study, we investigate how Mec1 and Rad53 control protein localization dynamics. We employ a high-throughput confocal imaging platform to monitor subcellular location changes of 322 proteins known to relocalize during MMS or HU treatment. We find that checkpoint signaling is responsible for 159 (49%) of the protein localization changes observed. Our analyses also reveal an unexpected subset of proteins whose proper subcellular localization depends on Rad53 yet is independent of the known canonical upstream activators of Rad53. Finally, we demonstrate that Rtg3 promotes phosphorylation of Rad53 during replication stress, revealing a biological connection between retrograde signaling and the DNA replication stress checkpoint response.

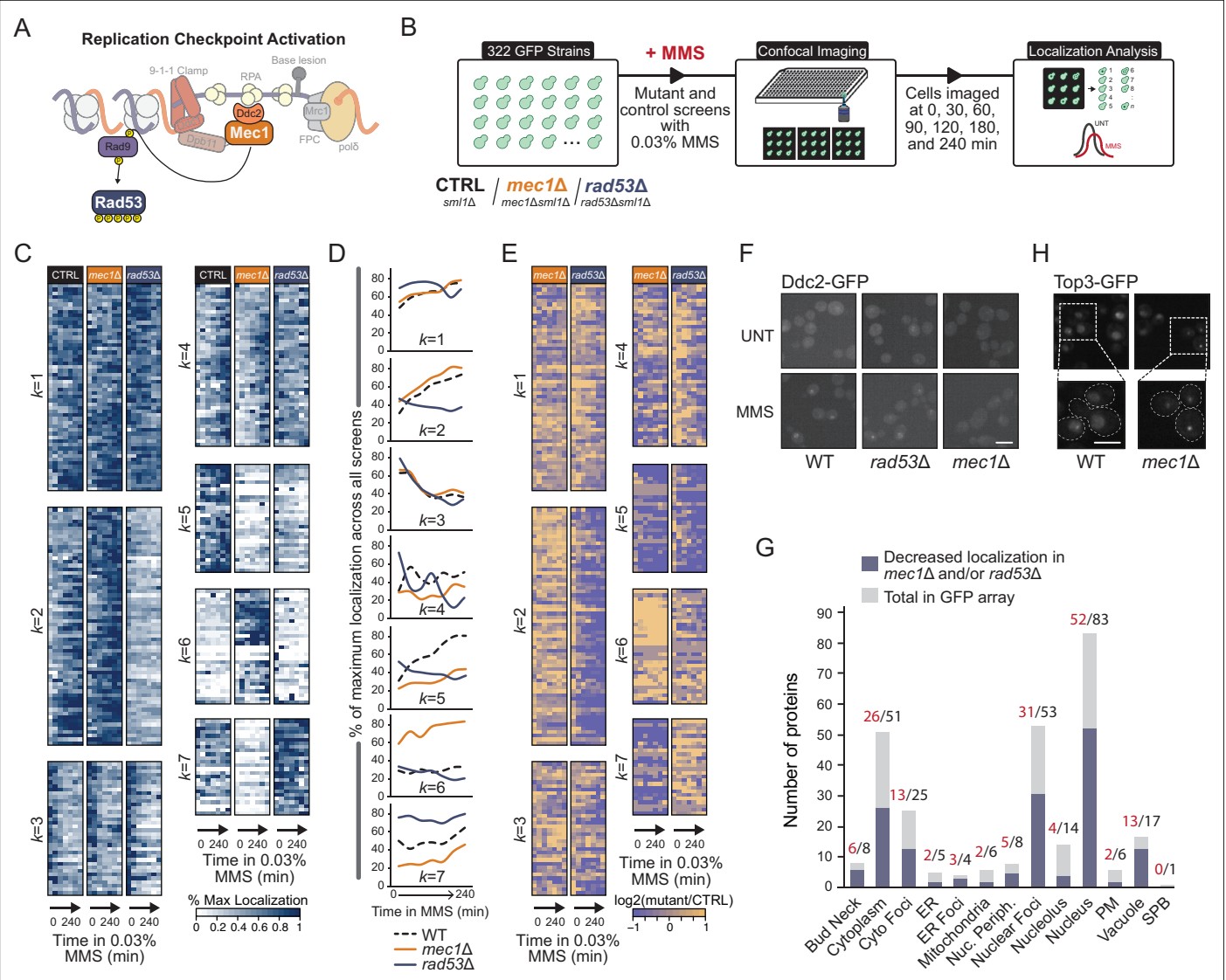

**Figure 1.** Checkpoint kinases Mec1 and Rad53 regulate protein re-localization. (**A**) A simplified schematic of replication checkpoint activation in yeast. (**B**) Outline of the imaging and quantification pipeline used in this study. *sml1Δ*, *mec1Δ sml1Δ*, or *rad53Δ sml1Δ* deletions were introduced into 322 strains, each expressing a unique protein-GFP fusion. Strains were treated with 0.03% methyl methanesulfonate (MMS) and imaged at seven times over the course of 240 min. Resulting images were subsequently segmented, and cells with changes in protein localization were identified. (**C**) Heatmaps of the percent of maximum protein re-localization in MMS (% max) for the control, *mec1Δ*, and *rad53Δ* screens. The resulting localization vectors were clustered into seven groups by *k*-means, with the *k*-group number indicated to the left of the heatmap. (**D**) Line plots of the mean extent of protein re-localization for each mutant screen at each time point for all proteins within each *k*-group. (**E**) The log2 ratio of the re-localization in mutant vs. control is displayed for each timepoint. (**F**) Representative images of Ddc2-GFP nuclear focus formation in WT, *mec1Δ*, and *rad53Δ* cells after 2 hr in 0.03% MMS. Scale bar is 5 µm. (**G**) The total number of proteins in each category of protein re-localization for all proteins assessed is shown in gray. The number of proteins with evidence of decreased protein re-localization rates in either *mec1Δ* or *rad53Δ* cells is shown in blue. (**H**) Representative images of Top3-GFP nuclear foci in wild type or *mec1Δ* cells in unperturbed conditions. Scale bar is 8 µm.

## Results

### Checkpoint kinases regulate replication stress-induced protein localization changes

Mec1 and Rad53 are essential checkpoint kinases that initiate and propagate the replication checkpoint signal to effector proteins via phosphorylation, with Rad53 directly phosphorylated by Mec1 (*Figure 1A*). Previous studies have identified 322 proteins that change subcellular location following replication stress, relocalizing with varying dynamics following MMS and HU treatment (*Chong et al.,*

*2015*; *Dénervaud et al., 2013*; *Mazumder et al., 2013*; *Srikumar et al., 2013*; *Tkach et al., 2012*; *Supplementary file 1*). Considering that these proteins are enriched for MMS-dependent phosphorylation targets, we assessed the role of the essential checkpoint kinases on their re-localization. We used Synthetic Genetic Array (SGA) to introduce *MEC1* or *RAD53* null mutations, in addition to an *SML1* deletion required for cell survival, into an array of 322 strains, each expressing one of the GFP-protein fusions previously found to relocalize during replication stress. The resulting strains were imaged by high-throughput confocal microscopy and protein localization was computationally quantified as previously described (*Ho et al., 2022*). Briefly, cells were grown to mid-logarithmic phase and imaged over four hours following 0.03% MMS treatment (*Figure 1B*). For each time point, the proportion of cells in the population with a protein-GFP localization change after MMS treatment was calculated using a computational image analysis pipeline (*Ho et al., 2022*; *Supplementary file 2*). Importantly, the analysis method includes normalization to the median cellular GFP signal to minimize the influence of protein abundance changes on identification of localization change events. To facilitate comparisons among the control and the checkpoint mutants, the percent maximum value (% max) was calculated, normalizing each timepoint to the maximum percent of cells for a given protein re-localization across the three screens (*Figure 1C and D*, *Supplementary file 3*). We also compared each mutant screen to the control screen by calculating $\log_2$ ratios of localization percentage values (% max) for each time point (*Figure 1E*). To reveal proteins with similar re-localization patterns, we clustered the resulting measurements for all three screens together using a *k*-means algorithm such that proteins with similar localization trajectories were grouped together. The analysis converged on seven *k*-means clusters (*Figure 1C, D and E*).

Upon calculation of the average localization change over time within each *k*-cluster, distinct patterns were evident. The MMS-induced re-localizations of the 40 proteins in cluster *k* = 5 depends on both *MEC1* and *RAD53* (*Figure 1C–E*), indicating that those proteins are downstream of the replication stress signal transduced by Mec1 to Rad53 to target protein. The 39 proteins in cluster *k* = 7 showed decreased re-localization only when *MEC1* was deleted, suggesting that these proteins are regulated by Mec1 but not by Rad53. As an example, the localization of Ddc2 to nuclear foci depends on *MEC1* but not on *RAD53* (*Figure 1F*), consistent with previous analysis indicating that Ddc2 localization is regulated by Mec1 and not by Rad53 (*Melo et al., 2001*). Two proteins (Rad50, Rfa1) in cluster *k* = 7 were identified as Mec1 targets in phosphoproteomic analyses (*Bastos de Oliveira et al., 2015*), and four (Mms21, Msh3, Nej1, Rad50) contain SQ/TQ motifs, which are consensus sites for Mec1 phosphorylation. Thus, cluster *k* = 7 contains proteins that are subject to Mec1 regulation, independent of Rad53. In total, 56% (159/284) of the proteins that change intracellular location (at least twofold change in percentage of cells with re-localization relative to wild-type at the corresponding time point) in response to MMS-induced DNA replication stress are regulated, either directly or indirectly, by the checkpoint kinase cascade in the expected fashion: their re-localization either depends on *MEC1* alone or depends on both *MEC1* and *RAD53*. We further noted that the proteins with reduced localization in checkpoint mutants are found in 12 of the 15 subcellular compartments represented in our study (*Figure 1G*). These data suggest that the replication checkpoint kinases regulate a global protein localization response following replication stress, with targets that are not solely restricted to the nuclear compartment.

We also noted that deletion of either kinase can result in proteins exhibiting increased re-localization, particularly in the absence of DNA replication stress (*Figure 1C–E*). For example, Top3 nuclear focus formation was increased when *MEC1* was deleted (*Figure 1H*). The 36 proteins in cluster *k* = 6, on average, show increased re-localization in *mec1Δ* even at t = 0. Similarly, the 39 proteins in cluster *k* = 7 show increased re-localization in *rad53Δ* that is evident in the absence of replication stress. A likely explanation for the increased spontaneous re-localization that we observe is that deletion of either kinase causes increased DNA replication stress, a phenomenon that has been documented extensively (*Brush et al., 1996*; *Chen and Kolodner, 1999*; *Craven et al., 2002*; *Hoch et al., 2013*; *Motegi et al., 2006*; *Myung and Kolodner, 2002*; *Pennaneach and Kolodner, 2004*; *Shimada et al., 2002*). Cells respond to genetically induced replication stress by activating the checkpoint response in the absence of the chemical stress inducer MMS. Ninety-six proteins show increased re-localization at t = 0 in a checkpoint kinase mutant, including a host of proteins that repair stressed and collapsed DNA replication forks, further emphasizing that DNA replication stress is present in checkpoint mutants and revealing a cohort of proteins that respond to the elevated stress.

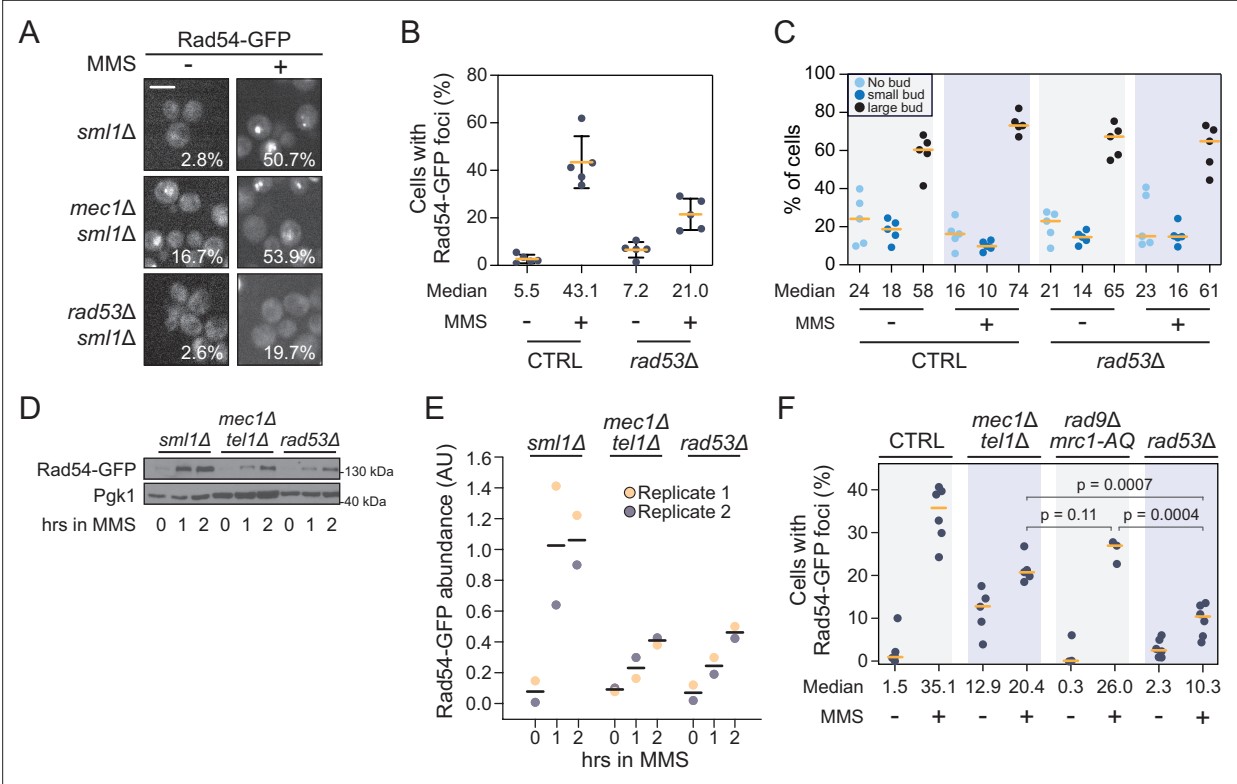

**Figure 2.** Rad54 nuclear focus formation requires Rad53 but not canonical checkpoint signaling. (**A**) Representative images of Rad54-GFP focus formation in *sml1Δ*, *mec1Δ sml1Δ*, or *rad53Δ sml1Δ* cells treated with 0.03% methyl methanesulfonate (MMS) for 2 hr. The percent of cells with Rad54-GFP foci is indicated. Scale bar is 5 μm. (**B**) The percent of cells with at least one Rad54-GFP nuclear focus, either untreated or after 2 hr of 0.03% MMS treatment, is plotted for wild type and *rad53Δ*. The median percent of cells with Rad54-GFP foci is indicated (n = 5). Error bars represent +/- one standard deviation. (**C**) The percent of unbudded, small-budded, and large-budded cells is plotted for wild type and *rad53Δ*, for untreated and MMS-treated cells. The median percent of cells for each category is indicated (n = 5). (**D**) Rad54-GFP was expressed from the *RAD54* promoter in the indicated checkpoint mutant strains. Mid-logarithmic phase cells were fixed at the indicated times following 0.03% MMS exposure. Cells were lysed and immunoblotted with anti-GFP or anti-Pgk1. (**E**) Densitometric quantification of Rad54-GFP protein in panel (**D**) and its replicate. Values are normalized to the Pgk1 loading control and expressed in arbitrary units (AU). Averages are indicated by the black horizontal bars. (**F**) Quantification of the percent of cells with at least one Rad54-GFP focus in each of the indicated strains, before and after 0.03% MMS treatment for 2 hr. The median percent is indicated (n ≥ 3). p-Values from a two-tailed *t*-test are indicated. For all plots, the orange horizontal bars indicate the medians.

The online version of this article includes the following source data and figure supplement(s) for figure 2:

**Source data 1.** Original and labeled immunoblot images for *Figure 2D*.

**Figure supplement 1.** Replicate of the immunoblot in *Figure 2D*.

## Rad53 functions during replication stress when Mec1 is absent

An unexpected and striking category of re-localization involved the 52 proteins in cluster *k* = 2 (*Figure 1C*), which displayed reduced re-localization when Rad53 is absent, but were largely unaffected by *mec1Δ*. Mec1 has long been recognized as the predominant upstream activator of Rad53. Studies in vitro have unambiguously shown that Mec1 phosphorylates Rad53 and promotes the trans-autophosphorylation of Rad53 (*Pellicioli et al., 1999*; *Sweeney et al., 2005*). In the absence of Mec1, activation of Rad53 by replication stress or DNA damage is not detected by standard protein kinase assays or by the characteristic mobility shift on SDS-PAGE that accompanies Rad53 activation. Yet our data indicate that a large number of protein re-localization events depend on *RAD53*, and on the presence of DNA replication stress, but apparently not on *MEC1*.

One example of a *RAD53*-dependent, *MEC1*-independent re-localization is shown in *Figure 2A*. The recombination repair protein Rad54 forms nuclear foci in response to replication stress. The fraction of cells displaying Rad54 in nuclear foci decreases in *rad53Δ* cells, but not in *mec1Δ* cells (*Figure 2A and B*). In subsequent assays, we use Rad54 re-localization as an exemplar of Mec1-independent regulation by Rad53. Rad54 foci form in MMS-induced replication stress and are easily

distinguishable and readily quantifiable. Furthermore, we were interested in Rad54 since it has clear connections to DNA damage and repair, playing an important functional role in homologous recombination (HR), where multiprotein complexes assemble into nuclear foci (*Lisby et al., 2004*). We first set out to establish that decreased Rad54 re-localization in *rad53Δ* is not due to cell cycle effects or to Rad54 abundance changes.

Cell cycle checkpoints arrest cells in G2-M in the presence of MMS to provide cells time to repair DNA damage (*Kupiec and Simchen, 1985*; *Siede, 1995*). Given that Rad53 is important for cell cycle arrest, it is formally possible that decreased Rad54 focus formation could be due to abnormal cell cycle progression in *rad53Δ*. It is well documented that *rad53Δ* mutants progress more rapidly through S phase relative to wild-type cells in conditions of replication stress (*McClure and Diffley, 2021*; *Tercero et al., 2003*). Importantly, the decrease in Rad54 foci in *rad53Δ* is clear at 2 hr, a time when both *rad53Δ* and wild-type cells display similar proportions of large-budded cells (*Figure 2C*). Therefore, it is unlikely that the decrease in Rad54 foci is due to cell cycle effects.

Replication stress activates transcriptional and translational programs, ultimately modulating protein levels to facilitate DNA synthesis and repair (*Jaehnig et al., 2013*; *Workman et al., 2006*). We tested whether Rad54 protein levels are reduced in *rad53Δ* cells after MMS treatment, which could explain the decrease in nuclear Rad54 foci. We found that Rad54 levels increase during MMS treatment, and that Rad54 abundance is lower in the checkpoint mutants than in the *sml1Δ* control strain (*Figure 2D*, *Figure 2—figure supplement 1*). However, Rad54 levels were similar in *mec1Δ tel1Δ sml1Δ* and *rad53Δ sml1Δ* cells (*Figure 2E*), indicating that the decreased Rad54 re-localization in *rad53Δ sml1Δ* compared to *mec1Δ tel1Δ sml1Δ* was not due to differences in Rad54 levels. We conclude that Rad54 focus formation is not strictly linked to Rad54 abundance.

Another potential explanation for the absence of *MEC1* dependence for Rad54 re-localization is that yeast carry a *MEC1* ortholog, *TEL1*, that in some instances can provide functional redundancy in activating Rad53 (*de la Torre-Ruiz et al., 1998*; *Morrow et al., 1995*; *Vialard et al., 1998*). We asked whether Rad53 still promotes Rad54 focus formation in cells with null mutations in both *MEC1* and *TEL1*. Cells expressing Rad54-GFP from the native *RAD54* locus were treated with 0.03% MMS for 2 hr and Rad54 nuclear focus formation was measured (*Figure 2G*). We detected a decrease in Rad54 focus formation in *mec1Δtel1Δ* compared to wild-type cells, but deletion of *RAD53* caused a greater decrease, indicating that in the absence of *MEC1* and *TEL1*, *RAD53* makes independent contributions to Rad54 re-localization (*Figure 2G*). Assessing Rad54 foci in *mec1Δ tel1Δ* was confounded by severe growth defects and morphological abnormalities, both of which could lead to an underestimate of re-localization. As an alternative approach, we mutated *MRC1* and *RAD9*, which encode the checkpoint mediators that propagate the replication stress signal from Mec1/Tel1 to Rad53 (*Figure 2G*). In the absence of mediator function the re-localization of Rad54 is unaffected, consistent with Rad54 re-localization being independent of signaling from Mec1/Tel1 to Rad53. Together, our data suggest that Rad54 focus formation depends on a pathway that transduces the signal elicited by MMS treatment to the checkpoint kinase Rad53 without a requirement for Mec1 and Tel1.

## Rad53 kinase activity is required for Rad54 focus formation

We next considered the domains of Rad53 that could promote protein re-localization without activation by Mec1/Tel1 and the mediators Mrc1 and Rad9. Since canonical activation of Rad53 was not essential for Rad54 re-localization, it was possible that a kinase-independent function of Rad53 was involved. In addition to its kinase domain, Rad53 contains two forkhead-associated domains at its N- and C-terminal ends, termed FHA1 and FHA2 (*Bashkirov et al., 2003*; *Durocher et al., 1999*; *Liao et al., 1999*). FHA1 functions in binding some Rad53 substrates, and FHA2 is important for interaction with Rad9 and subsequent Rad53 activation by trans-autophosphorylation (*Durocher et al., 2000*; *Durocher et al., 1999*; *Liao et al., 1999*; *Matthews et al., 2014*; *Pike et al., 2003*). We quantified the percentage of cells with Rad54-GFP nuclear foci in cells expressing inactivating mutations in each of the domains (R70A, R605A, and K227A/D319A/D339A [KD], located in the FHA1, FHA2, and kinase domains, respectively). Only the inactivation of the kinase domain (*RAD53KD*) resulted in decreased Rad54 re-localization to the level seen in *rad53Δ* cells (*Figure 3A*). Interestingly, mutation of FHA2 had only a modest effect, again consistent with the canonical Mec1-mediator axis of Rad53 activation playing at best a minor role in Rad54 re-localization. These results indicate that Rad53 kinase activity

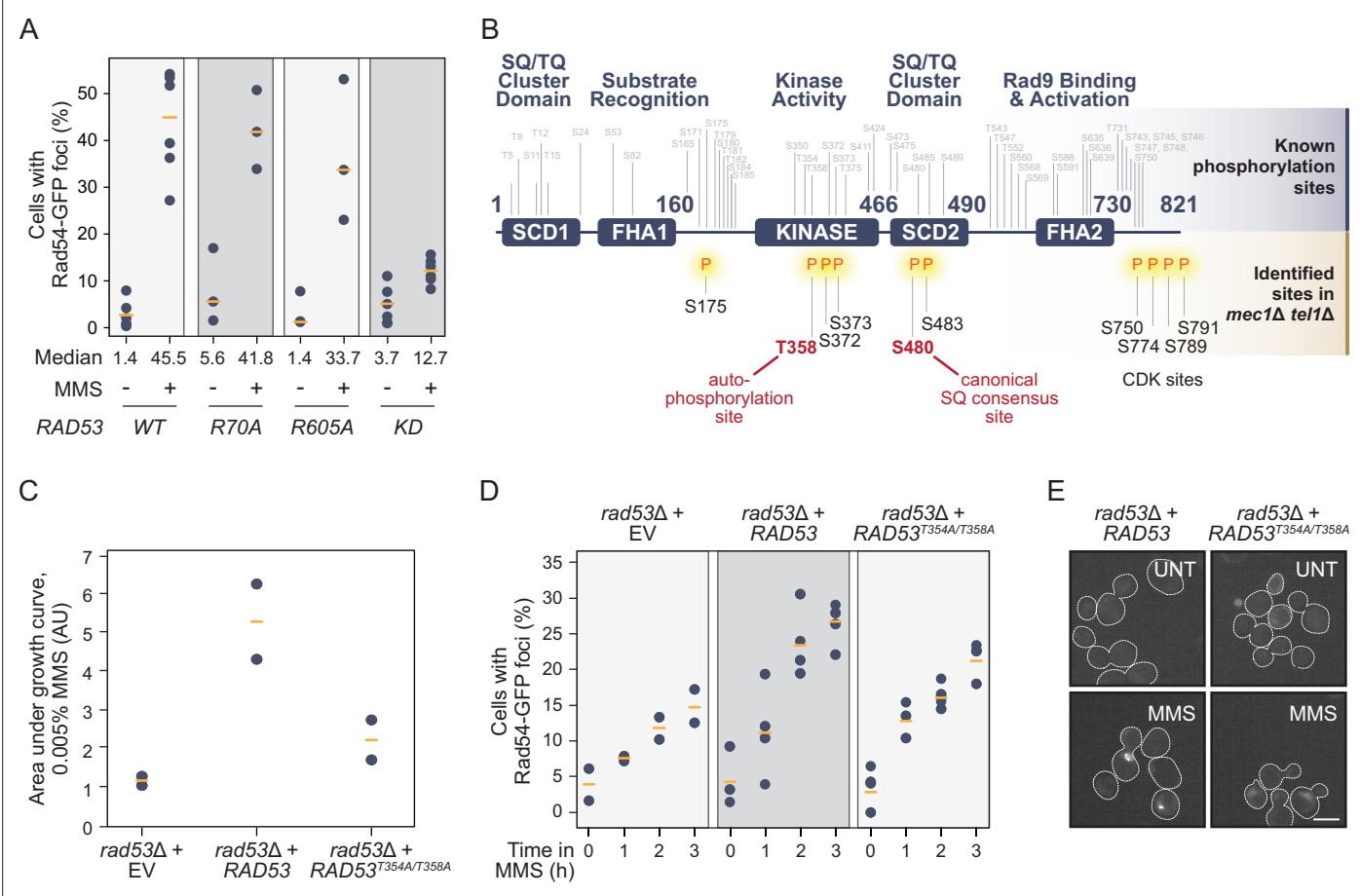

**Figure 3.** Rad53 is phosphorylated independent of Mec1/Tel1 and its kinase activity promotes Rad54 re-localization. (**A**) The percent of cells with at least one Rad54-GFP nuclear focus is plotted for the strains expressing the indicated *RAD53* alleles (KD = kinase dead = R70A, R605A), before and after 0.03% methyl methanesulfonate (MMS) treatment for 2 hr. The medians are indicated by the orange bars (n ≥ 3). (**B**) Schematic of the domain architecture of the Rad53 protein, with known phosphorylation sites annotated above in gray. Phosphorylation sites in *mec1Δ tel1Δ* cells identified by mass spectrometry are annotated below the schematic in black, with the two highlighted in red representing functional phosphorylation sites described in the literature. (**C**) The growth of *rad53Δ* strains expressing the indicated *RAD53* alleles (EV = empty vector) was measured and the area under each growth curve is plotted (AU = arbitrary units). Orange bars indicate the medians (n = 2). (**D**) The percent of cells with at least one Rad54-GFP nuclear focus is plotted for *rad53Δ* strains expressing the indicated *RAD53* alleles over 3 hr of 0.03% methyl methanesulfonate (MMS) treatment. Orange bars indicate the medians (n ≥ 2). (**E**) Representative fluorescence microscopic images of Rad54-GFP foci in *rad53Δ* strains expressing the indicated *RAD53* alleles. Scale bar is 5 μm.

is necessary for Rad54 re-localization in response to DNA replication stress, whereas neither FHA domain plays a substantial role.

Rad53 is normally activated following priming phosphorylation by Mec1, which promotes Rad53 dimerization and trans-autophosphorylation (*Gilbert et al., 2001*; *Ma and Stern, 2008*; *Pellicioli and Foiani, 2005*; *Wybenga-Groot et al., 2014*). Given that Rad53 kinase activity is tightly coupled to its phosphorylation, and that we find evidence for Rad53 activity outside of the characterized Mec1 pathway, we tested whether Rad53 was phosphorylated in *mec1Δ tel1Δ* cells. Rad53 fused with a 10X-FLAG epitope was expressed from its native locus, and cells were treated with MMS. Following affinity purification and phosphopeptide enrichment, Rad53 phosphopeptides were identified by mass spectrometry (*Figure 3B*, *Supplementary file 4*). We found only 10 phosphorylated residues in *mec1Δtel1Δ* cells, which is in stark contrast to the 48 sites on Rad53 normally occupied by phosphorylation in DNA damage conditions. Even in the absence of Mec1 and Tel1, phosphorylation was detected within the kinase and SCD2 domains of Rad53, including serine 480, which is a Mec1 S/T-Q consensus site in the SCD2 domain that promotes Rad53 activity when phosphorylated (*Lee et al., 2003*). Our data show that S480 can be phosphorylated by a kinase distinct from Mec1 and Tel1. Of

particular interest, we detected phosphorylation of T358, which is a trans-autophosphorylation site important for full activation of Rad53 (*Fiorani et al., 2008*; *Ma and Stern, 2008*). Phosphorylation of T358 suggests that Rad53 dimerization and trans-autophosphorylation have occurred, resulting in Rad53 kinase activation. We mutated T358 and the adjacent autophosphorylation site at T354 to alanine (*RAD53^{TA}* mutant) and found that it sensitized cells to MMS, and reduced Rad54 re-localization (*Figure 3C–E*). Taken together, these results indicate that Rad53 can be phosphorylated in response to DNA replication stress in a Mec1- and Tel1-independent manner, and demonstrate the important role of the Rad53 kinase activity in Rad54 re-localization despite the lack of a strong requirement for Mec1/Tel1 or mediator proteins.

## Retrograde signaling factor Rtg3 promotes Mec1-independent Rad54 focus formation

Our data suggest that Rad53 can respond to MMS, undergo dimerization, autophosphorylate, and promote proper localization of Rad54, all independent of Mec1 and Tel1 function. Therefore, we sought to identify the molecular factors involved in the non-canonical mode of Rad53 activation. Given that the replication checkpoint is a phosphorylation signaling cascade, and Rad53 itself is phosphorylated in MMS, we focused on protein kinases and kinase-related genes. We catalogued genes annotated in the *Saccharomyces* Genome Database (https://www.yeastgenome.org/) as either possessing or promoting kinase or phosphatase activity. We generated an array of 296 strains, each expressing Rad54-GFP from the *RAD54* locus and a unique kinase-related gene deletion or temperature sensitive allele (*Supplementary file 5*). To focus on Mec1-independent Rad53 activity, we also deleted *MEC1* in these strains (*Figure 4A*). We measured the percentage of cells with a change in Rad54 localization after 2 hr of MMS treatment (*Figure 4B*, *Supplementary file 5*) and identified 31 gene mutants that had decreased Rad54 foci (p<0.05, Student's *t*-test). The top three genes identified in our screen that resulted in reduced Rad54 focus formation are all part of the retrograde (RTG) signaling pathway (*RTG2*, *RTG3*, and *MKS1*). The RTG pathway regulates the expression of genes involved in metabolism and mitochondrial maintenance (*Jazwinski and Kriete, 2012*; *Jia et al., 1997*; *Komeili et al., 2000*; *Liao and Butow, 1993*; *Ruiz-Roig et al., 2012*), but has no known role in checkpoint signaling. To validate the screen results, we reconstructed *rtg2Δ mec1Δ* and *rtg3Δ mec1Δ* and assessed Rad54 focus formation induced by MMS (*Figure 4C and D*). Both strains displayed reduced levels of Rad54 re-localization compared to the respective single mutants. In particular, *rtg2Δ mec1Δ* had low levels of Rad54 foci, similar to *rad53Δ* cells. By contrast, *rtg2Δ rad53Δ* and *rtg3Δ rad53Δ* mutants did not have lower Rad54-GFP foci levels than *rad53Δ* (*Figure 4E*). Taken together, these results provide evidence that the *RTG* genes and *RAD53* function within the same pathway to regulate Rad54 focus formation.

To complement our analysis of Rad54 focus formation, we asked whether the RTG pathway influences the re-localization kinetics of other proteins in the Rad53-dependent but Mec1-independent category (cluster *k* = 2 in *Figure 1C*). We examined the re-localization of seven proteins in cluster *k* = 2 that had a detectable localization in the control condition and spanned distinct biological functions (Atg29, Dbf4, Exo1, Msn1, Mre11, Rpn4, and Xbp1) in *rtg3Δ*. Five proteins (Atg29, Msn1, Mre11, Dbf4, and Xbp1) had reduced re-localization during MMS treatment when *RTG3* was deleted (*Figure 4F*). Thus, the RTG pathway promotes the Rad53-dependent localization changes of at least five additional proteins, suggesting that the RTG pathway activates Rad53 rather than acting directly on Rad54. We conclude that Rtg3 is a regulator of multiple proteins whose re-localization depends on Rad53, but not on Mec1.

## The retrograde signaling pathway responds to replication stress

If the RTG pathway is transducing a replication stress signal to Rad53, we reasoned that the RTG pathway should respond to replication stress. The heterodimeric Rtg1/Rtg3 transcription factor is the downstream effector of the RTG pathway (*Figure 5A*; *Workman et al., 2006*) and translocates from the cytoplasm to the nucleus when the pathway is activated (*Sekito et al., 2000*). Microscopic inspection of Rtg3-GFP revealed that Rtg3 translocates to the nucleus following MMS-induced replication stress and persists in the nucleus for longer when the MMS concentration is increased (*Figure 5B*). Therefore, Rtg3 responds in a quantifiable manner to DNA replication stress, a prerequisite if Rtg3 transmits a replication stress signal to Rad53.

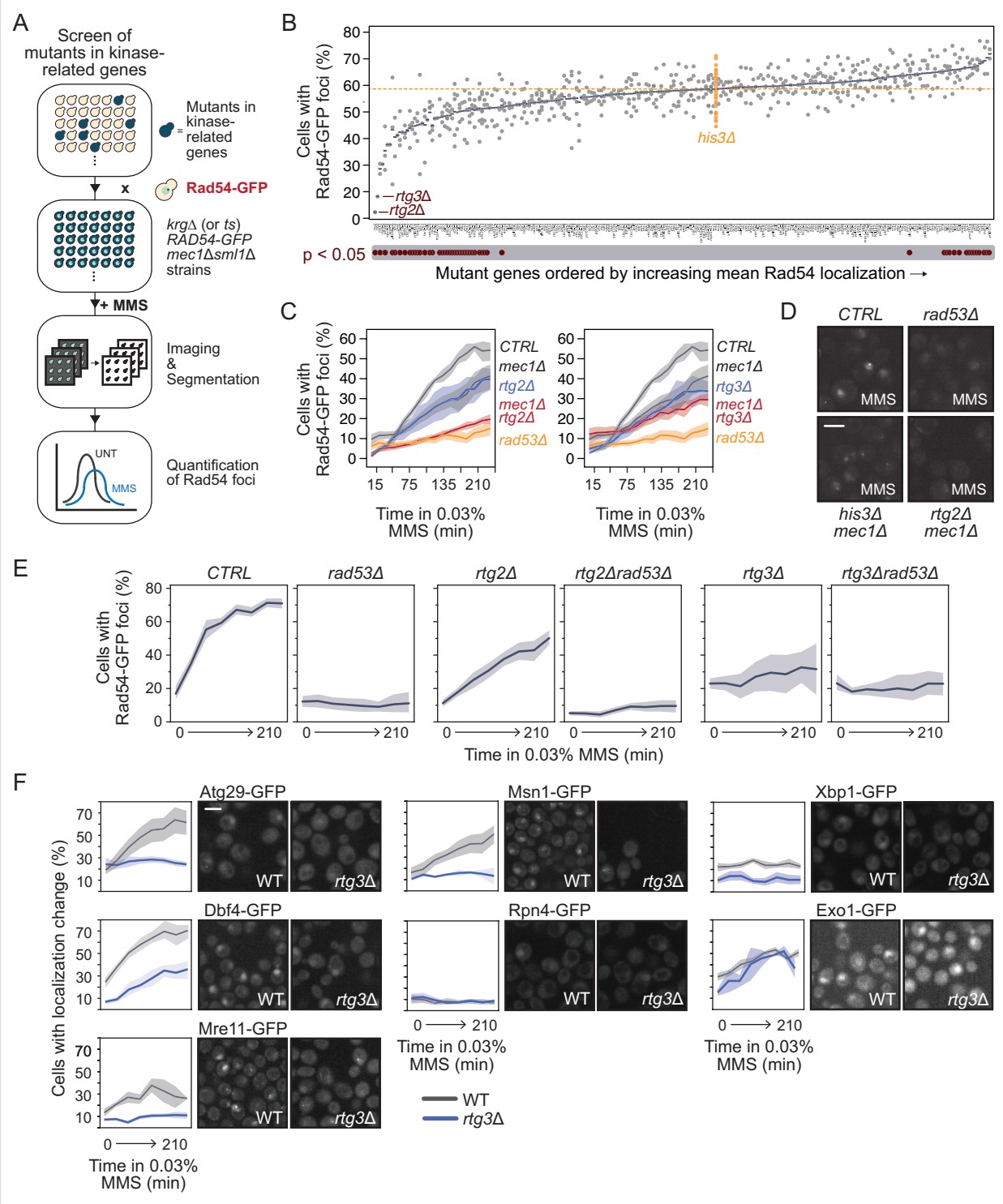

**Figure 4.** Retrograde signaling promotes Rad53-dependent Mec1-independent re-localizations. (**A**) Outline of the imaging and quantification pipeline for the small-scale screen. Briefly, *mec1Δ sml1Δ* cells expressing Rad54-GFP from its native locus were crossed with an array of unique strains each with a mutation in a kinase-related gene (*krg*). The resulting *krg^mut RAD54-GFP mec1Δ sml1Δ* strains were treated with methyl methanesulfonate (MMS) and imaged on a high-throughput confocal microscope. Resulting images were segmented and cells with changes in Rad54 localization were quantified. (**B**) The percent of cells with Rad54-GFP foci is plotted (gray circles, with blue bars indicating the means) for the indicated kinase-related gene mutants. The percent of cells with Rad54-GFP foci for 60 independent *his3Δ* strains is shown in orange, representing the wild-type control. Mutants with statistically supported (p<0.05) decreases or increases in Rad54-GFP foci compared to the *his3Δ* strains are indicated below the plot. The orange horizontal dotted line indicates the mean of the *his3Δ* strains. (**C**) The percent of cells with at least one Rad54-GFP nuclear focus is plotted for the indicated strains during

*Figure 4 continued on next page*

*Figure 4 continued*

0.03% MMS treatment for the indicated times. The double mutants (*rtg2Δ mec1Δ* and *rtg3Δ mec1Δ*) are shown in red. Orange bars indicate the medians (n = 3). (**D**) Representative images of Rad54-GFP foci in the indicated mutant strains. Scale bar is 5 μm. (**E**) The percent of cells with at least one Rad54-GFP nuclear focus is plotted for the indicated strains during 0.03% MMS treatment for the indicated times. Shaded regions indicate the 85% confidence interval. (**F**) *RTG3* was deleted in strains expressing the indicated protein-GFP fusions whose localizations were found to be *RAD53*-dependent and *MEC1*-independent. Strains were imaged on a confocal microscope during treatment with 0.03% MMS. The percent of cells with a protein localization change is plotted for WT (gray) and *rtg3Δ* (blue) for each protein-GFP strain (left panels; shaded regions indicate the 85% confidence interval). Representative images of strains expressing the indicated proteins after treatment with 0.03% MMS for 2 hr are shown to the right of each plot. Scale bar is 5 μm.

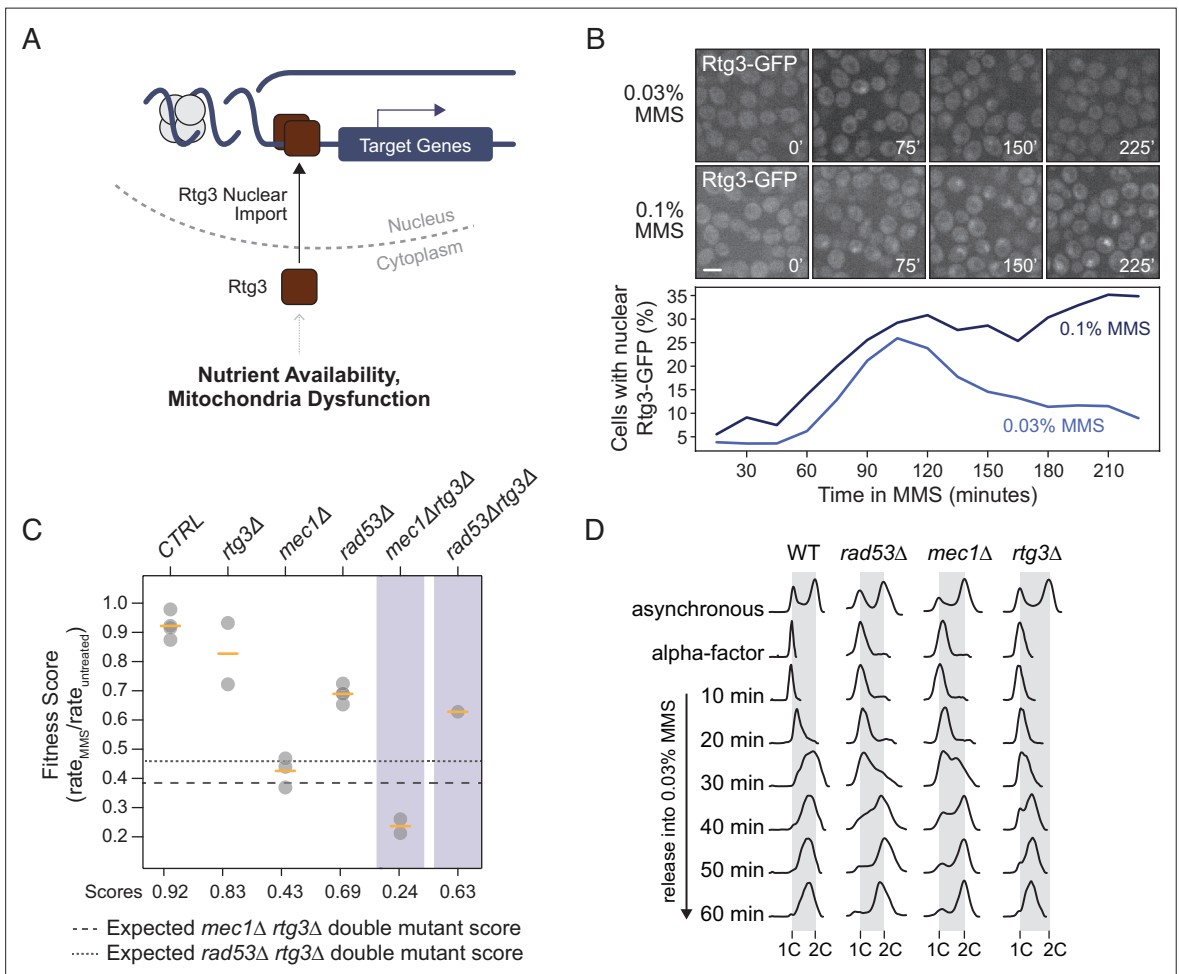

**Figure 5.** Rtg3 responds to DNA replication stress. (**A**) Under conditions of nutrient limitation or mitochondrial dysfunction, Rtg3 enters the nucleus and upregulates stress response genes. (**B**) Representative images of Rtg3-GFP localization at the indicated times during treatment with methyl methanesulfonate (MMS) (above). The percent of cells with nuclear Rtg-GFP is plotted at the indicated times during MMS treatment (below). Scale bar is 5 μm. (**C**) Fitness scores for the indicated strains grown in 0.005% YPD liquid culture were calculated as growth rate in MMS divided by growth rate in untreated conditions and plotted. Means are indicated by the orange bars (n ≥ 2). Expected double mutant fitness scores (multiplicative model) are indicated by the dashed lines. (**D**) Flow cytometric analysis of the indicated strains. Cells were arrested in G1 phase with alpha-factor and released into 0.03% MMS. Samples were fixed at the indicated times, DNA contents were measured, and plotted as histograms with events plotted on the y-axis and fluorescence plotted on the x-axis. Positions of 1C and 2C DNA contents are indicated.

The online version of this article includes the following figure supplement(s) for figure 5:

**Figure supplement 1.** Replicate of the cell cycle analysis in *Figure 5D*.

Deletion of *RTG1* or *RTG3* confers sensitivity to DNA replication stress induced by hydroxyurea (***Hartman, 2007***), and so we tested whether deletion of *RTG3* confers MMS sensitivity, and whether *RTG3* is in the same replication stress response pathway as *RAD53*. The *rtg3Δ* strain showed a small fitness defect in MMS (***Figure 5C***). When we combined *rtg3Δ* with *mec1Δ*, the fitness defect in MMS was greater than expected from analysis of the single mutants. By contrast, when we combined *rtg3Δ* with *rad53Δ* no greater-than-additive or greater-than-multiplicative fitness defect was evident, indicating that *RTG3* functions in MMS resistance in the same genetic pathway as *RAD53*, and parallel to *MEC1* (***Figure 5C***).

Since the checkpoint kinases are important in proper cell cycle progression and replication fork stabilization during replication stress (***Bermejo et al., 2011***; ***De Piccoli et al., 2012***; ***Iyer and Rhind, 2013***; ***Toledo et al., 2017***), we sought to directly assess S-phase progression in *rtg3Δ* cells. Cells were arrested in G1 phase and released synchronously into S phase in the presence of MMS (***Figure 5D***, ***Figure 5—figure supplement 1***). As expected, *mec1Δ* and *rad53Δ* cells exhibited checkpoint defects, both progressing faster than wild type cells and reaching 2C DNA content by 60 min (***Shimada et al., 2002***), whereas wild-type cells remain in S phase with less than 2C DNA contents. The *rtg3Δ* cells

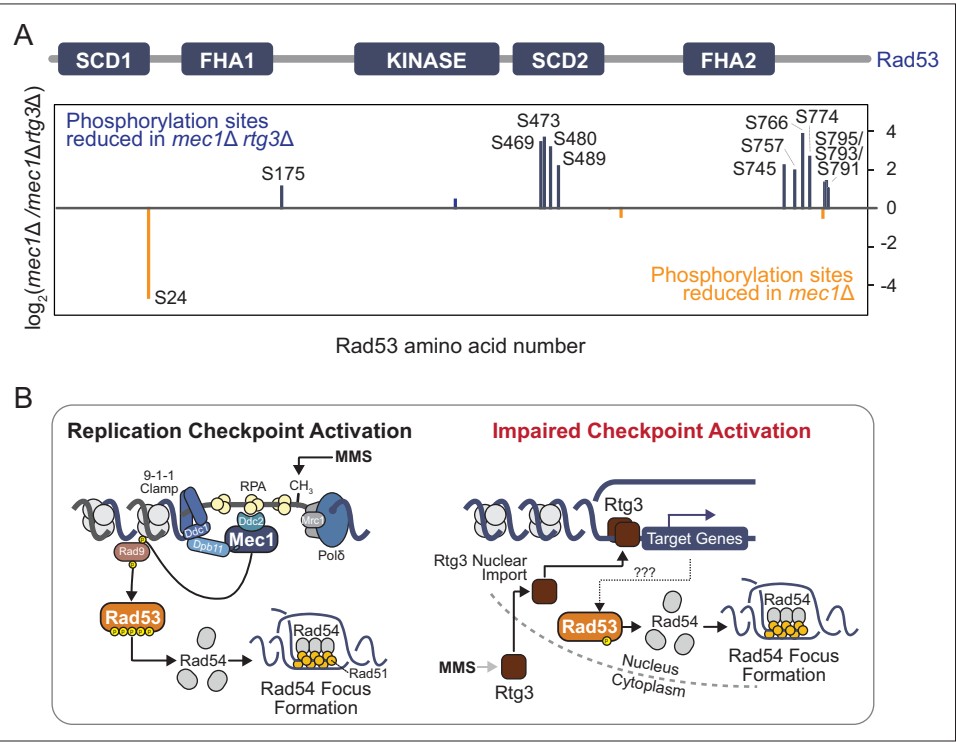

**Figure 6.** Rad53 phosphorylation is reduced when Rtg3 is absent. (**A**) Rad53-FLAG was immunoprecipitated from *mec1Δ sml1Δ* cells with or without *rtg3Δ*, phosphopeptides were enriched, and quantified by SILAC-MS. Phosphopeptides that are enriched or depleted in *mec1Δsml1Δrtg3Δ* cells are indicated in orange and blue, respectively. A schematic of the Rad53 protein and its domains indicates the location of the differentially phosphorylated sites. (**B**) A model for Rad53 activation in DNA replication stress when canonical checkpoint signaling is impaired in *mec1Δ* cells. Replication stress signals for the nuclear import of the Rtg3 transcription factor, resulting in gene expression changes. Target genes of Rtg3 promote Rad53 phosphorylation at functionally important residues, stimulating Rad53 activity, which is required for proper Rad54 focus formation.

The online version of this article includes the following source data and figure supplement(s) for figure 6:

**Figure supplement 1.** *rtg3* deletion does not alter the mobility shift of Rad53 observed in replication stress conditions.

**Figure supplement 1—source data 1.** Original and labeled immunoblot images for ***Figure 6—figure supplement 1***.

**Figure supplement 2.** Immunoprecipitation of Rad53-FLAG for SILAC-MS.

**Figure supplement 2—source data 1.** Original and labeled immunoblot images for ***Figure 6—figure supplement 2***.

progressed through S phase even more slowly than wild type. Although we cannot exclude some contribution of S phase entry delay, slow progression through S phase is consistent with the role of *RTG3* in replicating large replicons (*Theis et al., 2010*). We conclude that *RTG3* responds to DNA replication stress, promotes resistance to replication stress, functions within the *RAD53* pathway parallel to *MEC1*, and promotes S phase progression.

### Rtg3 promotes Rad53 phosphorylation in MMS in *mec1Δ* cells

If the RTG pathway is promoting signaling via Rad53 in the absence of Mec1, then we would expect that the activating phosphorylations that we identified on Rad53 (*Figure 3B*) might depend on Rtg3 function. Rad53 hyper-phosphorylation status did not appear to change in *rtg3Δ* cells, as assessed by mobility shift in immunoblot analysis (*Figure 6—figure supplement 1*). Therefore, to identify changes in phosphorylation at specific residues, we used a quantitative mass spectrometric approach to assess Rad53 phosphorylation status in *rtg3Δ mec1Δ* and *mec1Δ* cells (*Figure 6A*, *Figure 6—figure supplement 2*). We immunoprecipitated Rad53-FLAG from *rtg3Δ mec1Δ* and *mec1Δ* cells and found six sites where phosphorylation was reduced by at least two-fold in *rtg3Δmec1Δ* compared to *mec1Δ* (S469, S774, S473, S424, S766, and S743; *Figure 6A*, *Supplementary file 6*). An additional four sites were reduced by at least 1.5-fold (T563, S745, S489, and S475). Notably, three of these sites (S473, S480, and S489) are Mec1/Tel1 consensus phosphorylation sites, and S480 and S489 are located within the Rad53 SCD2 domain, which is important for Rad53 oligomerization and activation (*Ma and Stern, 2008*). Perhaps surprisingly, we did not detect changes in phosphorylation at T358. One possible reason may be the presence of redundant activities that promote T358 phosphorylation. Nevertheless, our data provide concrete evidence that the RTG pathway promotes phosphorylation of Rad53 on activation-relevant sites, in the absence of Mec1, consistent with the RTG pathway providing an unrecognized mode of Rad53 activation that can function in parallel to canonical Mec1-dependent Rad53 activation.

## Discussion

How the replication checkpoint regulates changes in protein subcellular location is largely unknown. By measuring replication stress-induced protein re-localizations in *mec1Δ* and *rad53Δ* cells, we showed that the checkpoint kinases coordinate a global protein re-localization response and regulate the movement of 159 proteins during replication stress. Most importantly, the proper localization of 52 proteins depends on Rad53, but not on the upstream kinases or mediator proteins of the replication checkpoint signaling cascade. We show that the RTG signaling pathway contributes to this novel mode of Rad53 activation. We propose a model (*Figure 6B*) where cells with an impaired replication checkpoint (i.e., *mec1Δ* cells) sense and respond to MMS-induced damage via the RTG pathway, resulting in nuclear accumulation of Rtg3. Since Rtg3 is a transcription factor, we postulate that Rtg3 regulates the expression of a kinase or kinases capable of phosphorylating Rad53. Alternatively, Rtg3 may function as a scaffold that promotes Rad53 dimerization and subsequent phosphorylation. The resulting phosphorylation of Rad53 appears to sustain at least a subset of Rad53 functions. How DNA replication stress is sensed by the RTG pathway, and the additional effectors that function in this pathway, remains to be discovered.

### Rad53 functions independently of canonical replication checkpoint signaling

Even though the replication checkpoint has been studied extensively, we suggest that Rad53 can be activated in the absence of its known activators, Mec1 and Tel1. Consistent with our inference, independent lines of evidence support the hypothesis that Rad53 exhibits genome stability functions independently of Mec1 and Tel1. In unperturbed cell cycles, *rad53Δ* cells display slower S-phase progression than *mec1Δ* (*Manfrini et al., 2012*), an effect that we also observed during replication stress (*Figure 5D*). Rad53 is recruited to replication fork proximal sites independently of the mediator Mrc1, suggestive of checkpoint-independent Rad53 recruitment (*Sheu et al., 2021*). Rad53 also binds to gene promoters independently of Mrc1 and Mec1, again indicating Rad53 function independent of Mec1 (*Sheu et al., 2021*). *MEC1* and *RAD53* have genetically distinct roles in stabilizing stressed replication forks (*Lanz et al., 2018*; *Segurado and Diffley, 2008*), although Mec1-independent

function of Rad53 was not noted. Finally, Rad53 function in the degradation of excess histones is Mec1-independent (*Gunjan and Verreault, 2003*). We demonstrated that Rad53 is phosphorylated in MMS-induced DNA replication stress in cells lacking Mec1 and Tel1. Two sites were of particular interest: (1) a canonical S/T-Q phosphorylation site at Ser480 in the SCD2 domain, known to be important for Rad53 oligomerization, and (2) an autophosphorylation site at Thr358 that, when mutated to alanine, severely compromises full Rad53 phosphorylation (*Chen et al., 2014*; *Ma and Stern, 2008*; *Wybenga-Groot et al., 2014*). Our data are consistent with a model in which Rad53 kinase can contribute to replication stress resistance even in the absence of its main activators, Mec1 and Tel1. However, it remains to be seen whether Mec1-independent activation of Rad53 is present constitutively or is only active when canonical checkpoint signaling is compromised. At present, we cannot distinguish between Mec1-independent activation of Rad53 being an acute response or a longer-term adaptation. We note, however, that the Mec1-independent Rad53 activation phenomenon is observed in independently constructed *mec1Δ* strains without a need for long-term passaging of cells and is observed in *rad9Δ* and *mrc1-AQ* contexts where cell growth is not compromised as it is in *mec1Δ*.

## Mec1-independent activation of Rad53 occurs through dimerization and phosphorylation

The typical mode for Rad53 activation requires its phosphorylation-dependent interaction with checkpoint mediator proteins Mrc1 or Rad9 (*Durocher et al., 2000*; *Sweeney et al., 2005*). Upon replication stress, Mec1 is recruited to and activated at RPA-coated ssDNA, where it then phosphorylates Rad9, among many other targets (*Rouse and Jackson, 2002*; *Zou and Elledge, 2003*). Mec1 phosphorylation enables Rad9 binding to the Rad53 FHA2 domain. Mec1 then phosphorylates several residues in the Rad53 SCD1 domain, resulting in the oligomerization, autophosphorylation, and full activation of Rad53 (*Sanchez et al., 1999*). Our data indicate that Mec1-independent activation of Rad53 occurs in a similar manner. The phosphorylation at Ser480 of Rad53 that we detect in the absence of Mec1 promotes oligomerization, and phosphorylation at Thr358 of Rad53 in the absence of Mec1 indicates that dimerization and trans-autophosphorylation has occurred. Since Rad9 and Mrc1 are not required for the Mec1-independent Rad53 activity that we observe, we propose that additional factors serve to increase the local concentration of Rad53 enough to promote oligomerization and trans-autophosphorylation. Concentration-dependent activation of Rad53 has been observed upon overexpression of Rad53 in bacteria (*Gilbert et al., 2001*), consistent with local concentration of Rad53 being sufficient to cause at least partial activation. Therefore, it is reasonable to speculate that a portion of total Rad53 protein can locally concentrate, activate, and exert its functions, independently of the canonical checkpoint activating factors Mec1, Tel1, Rad9, and Mrc1.

## Rtg3 functions to maintain genome stability

Changes in the mitochondrial state, or its dysfunction, activate a transcriptional program that adjusts metabolic activities and stress responses and is termed retrograde (RTG) regulation (*Giannattasio et al., 2005*; *Jazwinski, 2013*). In yeast, Rtg1 and Rtg3 are basic helix-loop-helix leucine zipper transcriptional activators that translocate into the nucleus when cells experience mitochondrial dysfunction. Rtg2 is another key protein component of the RTG signaling pathway and is required for Rtg1/3 nuclear accumulation. RTG signaling integrates with another signaling pathway that senses intracellular stress, the TOR pathway (*Crespo et al., 2002*). Unexpectedly, we found that RTG signaling is also involved in genome maintenance. Most strikingly, our finding that Rtg3 contributes to Rad53 phosphorylation status implicates retrograde response factors in replication checkpoint signaling, Rad53 function, and cellular fitness upon exposure to DNA replication stress.

Several lines of evidence support the notion that retrograde signaling impinges upon genome integrity pathways. Rtg2 inhibits the formation of extrachromosomal ribosomal DNA circles (ERC), a form of genome instability (*Borghouts et al., 2004*). ERCs arise from recombination events between distal sites within a chromosome and are self-replicating episomes of ribosomal DNA (*Stults et al., 2008*; *Szostak and Wu, 1980*). Accumulation of ERCs has been correlated with yeast aging, and circular DNAs capable of driving genetic heterogeneity and therapeutic resistance are abundant in certain human cancer cells (*Koche et al., 2020*; *Yan et al., 2020*). Our understanding of circular DNA

biogenesis, regulation, and function is far from complete. Nevertheless, the connection between Rtg2 and ERCs suggests roles for retrograde signaling in the maintenance of genome integrity.

Rtg3 also cooperates with Ino4 to regulate the expression of metabolism genes after MMS exposure (*Workman et al., 2006*). TCA cycle regulation, threonine biosynthesis, and permease trafficking pathways coordinate with one another to buffer dNTP pools, suggesting an indirect role of RTG in maintaining proper cellular nucleotide levels, which are crucial for normal DNA replication (*Koche et al., 2020*). Importantly, *rtg1, rtg3,* and *rtg3* deletion mutants exhibit severe growth defects in conditions of nucleotide limitation and have reduced levels of cellular dNTPs. In contrast, *RTG3* over-expression induces ribonucleotide reductase expression, further linking *RTG3* to dNTP pool regulation (our unpublished data). Consistent with a role in replication, Rtg3 was identified in a genome-wide search for factors that facilitate replication of long inter-origin gaps (*Theis et al., 2010*). Similarly, we found that Rtg3 promotes proper S-phase progression. Like RTG signaling, Rad53 has essential roles in elevating dNTP levels for repair, preventing late origin firing, and phosphorylating Mrc1 to antagonize CMG helicase unwinding and slow the replication fork (*Lopez-Mosqueda et al., 2010*; *McClure and Diffley, 2021*; *Morafraile et al., 2019*; *Morafraile et al., 2015*; *Szyjka et al., 2008*; *Tsaponina et al., 2011*). Since our data show that Rtg3 influences Rad53 phosphorylation status and function, we suggest that *RTG3* functions in the completion of DNA replication during replication stress, at least in part through Rad53 activation. Thus, in addition to identifying the complement of proteins whose location is regulated by cell cycle checkpoint kinases, we have found a previously uncharacterized mode of checkpoint signaling.

## Materials and methods

### Yeast strains

All yeast strains were in the S288C (BY) background. Unless otherwise stated, all strains with either a *MEC1* or *RAD53* deletion also contain an *SML1* deletion. Strains were constructed using genetic crosses and standard PCR-based gene disruption and epitope tagging techniques.

### Generating checkpoint mutant GFP collections by synthetic genetic array

An array comprised of 322 yeast strains was created, each consisting of unique ORF tagged with GFP and the following genotype: *MATa XXX-GFP::HIS3MX his3Δ1 leu2Δ0 met15Δ0 ura3Δ0*. Three query strains were constructed: (1) *mec1Δ::kanMX sml1Δ::hphMX*, (2) *rad53Δ::NATMX sml1Δ::hphMX,* and (3) *sml1Δ::hphMX,* each with the genotype *MATα can1pr-RPL39pr-tdTomato::CaURA3::can1Δ::STE2 pr-LEU2 his3Δ1 leu2Δ0 lyp1Δ0*. Synthetic genetic array was performed as described (*Baryshnikova et al., 2010*; *Tong et al., 2001*) to introduce the gene deletions from the query strains into the GFP array of 322 strains. The final strains were selected on solid agar media SD/MSG – his – leu -lys – arg + 50 µg/mL canavanine +50 µg/mL thialysine + 300 µg/mL hygromycin. The *MEC1* and *RAD53* deletion strains were selected on the same media with the addition of 160 µg/mL geneticin and 100 µg/mL nourseothricin, respectively.

### High-throughput confocal microscopy

For protein localization screens involving the 322 GFP mini array, cells were grown to logarithmic phase at 30°C and imaged by GFP fluorescence microscopy using the OPERA High Content Screening System (PerkinElmer), as described (*Torres and Brown, 2015*). Briefly, cells were grown in low fluorescence media (LFM – 1.7 g/L yeast nitrogen base without amino acids and without ammonium sulfate, 2% glucose, 1× methionine, 1× uracil, 1× histidine, and 1× leucine) in 96-well polypropylene U-bottom culture boxes to logarithmic phase at 30°C. Cells were then diluted to an $OD_{600}$ = ~0.02–0.05 in 384-well, clear-bottom plates for microscopic imaging (PerkinElmer 6007550) and incubated at 30°C for 1 hr to allow cells to settle. The cells were imaged on the PerkinElmer OPERA imaging system before and after MMS treatment with the following configuration: primary dichroic filters reflecting excitation wavelengths of 488 nm and 561 nm, detection dichroic set to 568 nm, 520/35 nm emission filter for Camera 1, 600/35 nm emission filter for Camera 2, laser power set to 100%, and exposure for GFP and RFP channels at 800 ms.

For protein localization screens involving the kinase-related array, and GFP fluorescence assays involving Rad54-GFP, imaged on the PerkinElmer OPERA Phenix System. Cells were grown in synthetic complete (SC) media (6.7 g/L yeast nitrogen base without amino acids and with ammonium sulfate, 2% w/v glucose, 1× amino acids) at 30°C to mid-logarithmic phase ($OD_{600}$ = 0.3–0.7). Cells were diluted and prepared as above, treated with 0.03% MMS for the indicated timepoints, then imaged on the OPERA Phenix with the following configuration: primary dichroic filters reflecting excitation wavelengths of 488 nm and 561 nm, detection dichroic set to 568 nm, 520/35 nm emission filter for Camera 1, 600/35 nm emission filter for Camera 2, laser power set to 100%, and exposure for GFP and RFP channels at 800 ms.

## Calculating percentages of cells exhibiting a localization

For all images acquired from the OPERA and OPERA Phenix, image segmentation of single yeast cells was conducted using CellProfiler 3.1 (*McQuin et al., 2018*), GFP pixel intensity measurements of the resulting segmented single cells was performed using Python 3.7.0 (https://www.python.org), and data analyses and visualization were performed using Python and/or R (https://www.r-project.org). Briefly, for every cell in our screens, the 95th percentile of the GFP pixel intensity was calculated and normalized to the cell median GFP intensity, representing the 'localization value' of a cell. The normalization to the cellular median GFP intensity was performed to minimize changes in the localization value that occur as a result of increasing or decreasing protein abundance levels. For each unique protein-GFP fusion, the localization value was determined for untreated cells, and the median and median absolute deviation (MAD) for this distribution were calculated. For every timepoint before and after drug treatment, a cell with a localization value greater than or less than 1.5 MADs from the median was considered to a cell exhibiting a protein re-localization event. Calculation of the percent max value was performed as described in *Ho et al., 2022*. For the kinase-associated gene screen with cells expressing Rad54-GFP, the 98th percentile of GFP pixel intensity was used.

## Budding index and live-cell GFP fluorescence microscopy

Cells expressing Rad54-GFP were grown in synthetic complete (SC) media (6.7 g/L yeast nitrogenous base without amino acids and with ammonium sulfate, 2% glucose, and all amino acids supplemented) at 30°C to mid-logarithmic phase ($OD_{600}$ = 0.3–0.7). Cells were either left unperturbed or treated with 0.03% MMS for 120 min. Cells were imaged on a Nikon Eclipse Series Ti-2 inverted widefield microscope using open software μ manager (https://micro-manager.org/). Budding index and the percentage of cells with a localization signal were manually assessed by visual inspection.

## Flow cytometry

For cell cycle analyses, 1 mL of cultures at $OD_{600}$ = 0.5 were collected at the indicated times, fixed in 70% ethanol, and stored at 4°C until sample processing. Cells were washed in $ddH_2O$, resuspended in 0.5 mL of 50 mM Tris-Cl (pH 8.0) with 2 mg/mL RNase A, and incubated for 2 hr at 37°C. Cells were then pelleted and resuspended in 0.5 mL of 50 mM Tris-Cl (pH 7.5) with 1 mg/mL proteinase K (BioShop PRK403) and incubated for 1 hr at 50°C. Cells were pelleted and resuspended in 200 mM Tris-Cl (pH 7.5), 200 mM NaCl, and 78 mM $MgCl_2$, and stored at 4°C. Immediately before analyzing samples on the flow cytometer, 0.1 mL of sample was added to 0.5 mL SYBR green solution (1:5000 dilution, Sigma-Aldrich S9430), sonicated briefly, and analyzed using a FACS Canto II (Becton Dickinson). A total of 10,000 events were collected, and plots were generated using FlowJo software version 10.0.8.

## Rad53 phosphorylation mapping

Yeast expressing *RAD53-FLAG* tagged at its native locus were grown to mid-log phase in YEPD medium and treated with 0.03% methyl methanesulfonate for 2 hr. Pellets were lysed by bead-beating with 0.5 mm glass beads for three cycles of 10 min with a 1 min rest between cycles at 4°C in lysis buffer (150 mM NaCl, 50 mM Tris pH 7.5, 5 mM EDTA, 0.2% NP40) supplemented with complete EDTA-free protease inhibitor cocktail (Roche), 5 mM NaF, and 10 mM β-glycerophosphate. RAD53-FLAG was immunoprecipitated from approximately 5 mg of protein extract with FLAG antibody-conjugated agarose resin and eluted with 1% SDS in 100 mM Tris pH 8.0. Eluate was reduced using 10 mM DTT and subsequently alkylated with 25 mM iodoacetamide followed by precipitation for 1 hr in PPT

solution (50% acetone, 49.9% ethanol, 0.1% acetic acid) on ice. Cells were washed once with PPT solution and then resuspended in urea/tris solution (8 M urea, 50 mM Tris pH 8.0). 8 M urea containing solubilized pellet was diluted to 2 M urea using milliQ water and then digested overnight at 37°C with 1 ug of trypsin GOLD (Promega). Phosphopeptides were selectively enriched using a home-made Fe-NTA resin micro-column and then subjected to LC-MS/MS analysis on a Thermo-Fisher Q-Exactive HF mass spectrometer. Raw MS/MS spectra were searched using SORCERER (Sage N Research, Inc) running SEQUEST software over a composite *S. cerevisiae* peptide database consisting of normal protein sequences downloaded from the *Saccharomyces* Genome Database (SGD) and their reversed protein sequences as a decoy to estimate the false discovery rate (FDR) in the search results. Searching parameters included a semi-tryptic requirement, a mass accuracy of 15 ppm for the precursor ions, a static mass modification of 57.021465 daltons for alkylated cysteine residues, and a differential mass modification of 79.966331 daltons for phospho-serine, -threonine and -tyrosine residues.

## Whole-cell extracts and immunoblotting

Cells were cultured and grown to $OD_{600}$ = 0.3–0.5 and treated with 0.03% MMS for the indicated times. Cells were then diluted to $OD_{600}$ = 1.0 in 10% trichloroacetic acid for 15 min, shaking gently, then pelleted by centrifugation at 3000 rpm for 5 min. Next, cells were resuspended in 2× Laemmli sample buffer and lysed by vortex on maximum speed in the presence of 0.5 mm glass beads, at 4°C for 10 min. Samples were boiled at 95°C for 10 min and stored long term at −80°C. Proteins were resolved on SDS-PAGE gels and detected by immunoblotting with anti-Rad53 (1:5000, Abcam ab104232), anti-PGK (1:1 000 000, Novex 459250), anti-GFP (1:5000, ClonTech 632375), and anti-FLAG M2 (1:5000, Sigma F3165).

## Growth rate assays

Saturated cultures of yeast strains were diluted to an $OD_{600}$ = 0.05 in 100 μL of YPD (with or without the indicated concentrations of drug) in flat-bottom 96-well plates. Plates were inserted into a TECAN microplate analyzer set at 30°C with gentle agitation and the $OD_{600}$ was measured every 10 min for 48 hr. The growth rate (maximum slope) and the area under the curve (AUC) were calculated using an R package designed by Danielle Carpenter (available here).

## Computational analyses, data, and software availability

Statistical analysis, data manipulation, and data visualization were performed in R (https://www.r-project.org) or Python (https://www.python.org). All the details of the data analysis can be found in the 'Results' and 'Materials and methods' sections. Python and R scripts used for data processing, analysis, and visualization are available online on GitHub (https://github.com/bqho/localization-quantification copy archived at *Ho, 2023*). Complete and raw image data, and metadata files, for the control screen and checkpoint mutant screens are deposited at the Image Data Resource (IDR) (https://idr.openmicroscopy.org; accession number IDR0140 for control screen; accession number IDR0145 for *MEC1* and *RAD53* deletion mutant screens). All other relevant data are available from the authors without restriction.

# Acknowledgements

This work was supported by the Canadian Institutes of Health Research (FDN-159913 to GWB), the National Institutes of Health grant (R35GM141159 to MBS), an Ontario Government Scholarship, and a Natural Sciences and Engineering Research Council of Canada CGS-D award (to BH). GWB holds a Canada Research Chair (Tier 1). We are grateful to work on the lands of the Mississaugas of the Credit, the Anishnaabeg, the Haudenosaunee and the Wendat peoples, land that is now home to many diverse First Nations, Inuit, and Métis peoples.

## Additional information

### Funding

| Funder | Grant reference number | Author |
|---|---|---|
| Canadian Institutes of Health Research | FDN-159913 | Brandon Ho<br>Raphael Loll-Krippleber<br>Nikko P Torres<br>Grant W Brown |
| Natural Sciences and Engineering Research Council of Canada | CGS-D | Brandon Ho |
| National Institutes of Health | R35GM141159 | Ethan J Sanford<br>Marcus B Smolka |

The funders had no role in study design, data collection and interpretation, or the decision to submit the work for publication.

### Author contributions

Brandon Ho, Conceptualization, Resources, Software, Formal analysis, Investigation, Methodology, Writing - original draft, Writing - review and editing; Ethan J Sanford, Formal analysis, Investigation, Methodology, Writing - review and editing; Raphael Loll-Krippleber, Investigation, Visualization; Nikko P Torres, Investigation, Methodology, Writing - review and editing; Marcus B Smolka, Supervision, Funding acquisition, Writing - review and editing; Grant W Brown, Conceptualization, Supervision, Funding acquisition, Writing - original draft, Writing - review and editing

### Author ORCIDs

Brandon Ho ⬚ http://orcid.org/0000-0002-7566-3007
Marcus B Smolka ⬚ http://orcid.org/0000-0001-9952-2885
Grant W Brown ⬚ http://orcid.org/0000-0002-9002-5003

### Decision letter and Author response

Decision letter https://doi.org/10.7554/eLife.82483.sa1
Author response https://doi.org/10.7554/eLife.82483.sa2

## Additional files

### Supplementary files

• Supplementary file 1. Complete array of GFP-tagged proteins analyzed in this study.

• Supplementary file 2. Percent of cells with protein re-localization in the control and checkpoint mutant screens.

• Supplementary file 3. Cells with protein re-localization (percent of maximum) in the control and checkpoint mutant screens.

• Supplementary file 4. Phosphorylated Rad53 peptides identified in affinity-purified Rad53-FLAG protein from *mec1Δtel1Δ* cells.

• Supplementary file 5. Percentage of cells with Rad54-GFP re-localization in *mec1* deletion and kinase-related gene deletion double mutants.

• Supplementary file 6. Changes in Rad53 phosphorylation sites identified by SILAC mass spectrometry.

• MDAR checklist

### Data availability

Datasets are provided in Supplementary Files. Images were deposited to IDR microscopy open access under accession numbers IDR0140 and IDR0145. Source data files have been provided for Figures 2 and 6.Python and R scripts used for data processing, analysis, and visualization are available online on GitHub (https://github.com/bqho/localization-quantification, copy archived at *Ho, 2023*).

The following datasets were generated:

| Author(s) | Year | Dataset title | Dataset URL | Database and Identifier |
|---|---|---|---|---|
| Ho B, Loll-Krippleber R, Torres NP, Cuny A, Rudolf F, Brown GW | 2022 | Phenotypic Heterogeneity in the DNA Replication Stress Response Revealed by Quantitative Protein Dynamics Measurements | https://idr.openmicroscopy.org/webclient/?show=screen-3253 | Image Data Resource, IDR0140 |
| Ho B, Sanford EJ, Torres NP, Smolka MB, Brown GW | 2022 | Mec1-Independent Activation of the Rad53 Checkpoint Kinase Revealed by Quantitative Analysis of Protein Localization Dynamics | https://idr.openmicroscopy.org/webclient/?show=screen-3303 | Image Data Resource, IDR0145 |

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
