## [Editor Report]

This fundamental study identifies a novel non-canonical activation mode for the central checkpoint kinase Rad53, a phosphorylation event that does not depend on Mec1 and instead depends on proteins involved in retrograde signaling through Rtg3. The evidence supporting the conclusion is compelling, with rigorous genetic, imaging, and proteomic approaches. Collectively, the findings convincingly demonstrate unanticipated complexities in the cellular DNA replication stress response, which will be of broad interest to the genome stability field.

---

## [Decision Letter]

**Decision letter after peer review:**

Thank you for submitting your article "Mec1-Independent Activation of the Rad53 Checkpoint Kinase Revealed by Quantitative Analysis of Protein Localization Dynamics" for consideration by *eLife*. Your article has been reviewed by 3 peer reviewers, and the evaluation has been overseen by a Reviewing Editor and Jessica Tyler as the Senior Editor. The following individuals involved in review of your submission have agreed to reveal their identity: Boris Pfander (Reviewer #1); Dirk Remus (Reviewer #2).

The reviewers and the Reviewing Editor agree that the study is significant and well-conducted, and they support publication in *eLife*. The identification of pathways responsible for Rad53 activation independently from Mec1 is particularly relevant to the fields of DNA repair, checkpoint, and genome stability. The identification of a crosstalk between the retrograde pathway and checkpoint activation is also an interesting and important addition to these fields.

Essential revisions:

1. Did rtg3∆ and 2∆ affect any other HR proteins besides Rad54? (mostly Reviewer 3 but also pointed out by Reviewer 1)

2. The readout used in the work was GFP signal intensity and not subcellular localization. The two could be correlated in some but not all cases. For example, increase in GFP intensity could reflect protein forming DNA repair foci or protein level increase. It will help increase the clarity of the work if the readout can be better defined in the text and the distinction can be made regarding changes of signal intensity vs protein localization (Reviewer 3).

3. It can be difficult to discern whether the observed changes of GFP signal intensity in a mutant reflected a direct effect OR a secondary response to the chronical loss of the protein, particularly when some mutants used in the work showed chronic growth defects. It would be useful to validate the main findings using degron alleles. For example, does degron-driven loss of Mec1/Tel1 also affected Rad54 foci levels less prominently than Rad53 loss (Reviewer 3)?

4. Figure 4, did rtg2∆ rad53∆ and rtg3∆ rad53∆ cells show similar reduction of Rad54-GFP signals as rad53∆ cells? Could the authors speculate why rtg2∆ but not rtg3∆ worsened Rad54 signal reduction in mec1∆ cells? (Reviewer 3)

5. Rad53 phosphorylation have only been tested in mec1 background. It is important to show what is the contribution of Rtgs to Rad53 phosphorylation in a WT background (Reviewers 1 and 3).

6. Provide more timepoints for the FACS profiles to better highlight the effect of Rtg3 deletion in replication progression (Reviewer 1 and 2)

7. Provide biological duplicates for Figure 2D and 2E (Reviewer 3)

*Reviewer #1 (Recommendations for the authors):*

This study shows the power of yeast genetics when combined with modern quantitative imaging and mass spectrometry technology. I would ask the authors to address the following points prior to publication.

1. Rad53 phosphorylation have only been tested in mec1 background. I think it would be an important information to the reader to show what is the contribution of Rtgs in an otherwise WT background. Even if under these conditions Mec1-dependent phosphorylation would dominate that would be important information.

2. Despite the initial screening effort, the authors focus exclusively on Rad54 focus formation throughout the paper. Ideally, they would show some of the later findings of the paper for other factors to show universality of their findings. At least, however, they should discuss why they focused on Rad54 and what cellular activity they think Rad54 focus formation corresponds to.

3. Compared to the very high experimental standard throughout the paper, I was not too convinced by the assay measuring S phase progression and would suggest that it is repeated with more timepoints.

*Reviewer #2 (Recommendations for the authors):*

Figure 5C: The data for the control and RTG3-deletion background appear to be duplicated in the left and right panel of this figure. I recommend combining the two panels into a single panel to avoid such duplication. This should not diminish the clarity of the result presentation.

Figure 5D: The effect of RTG3 deletion on the FACS profiles is rather minor and may also be influenced by a delayed release from the α-factor arrest. How reproducible are these effects? Also: Since the FACS profiles of the DNA content are likely dominated by origin firing, have the authors tried to look for effects on origin activity after replication stress in these different genetic backgrounds? While not strictly required, this could provide further support for a role of Rtg3 in controlling S phase progression.

*Reviewer #3 (Recommendations for the authors):*

I have a few suggestions to help clarify methods used and data interpretation.

1) The readout used in the work was GFP signal intensity and not subcellular localization. The two could be correlated in some but not all cases. For example, increase in GFP intensity could reflect protein forming DNA repair foci or protein level increase. It will help increase the clarity of the work if the readout can be better defined in the text and the distinction can be made regarding changes of signal intensity vs protein localization.

2) It can be difficult to discern whether the observed changes of GFP signal intensity in a mutant reflected a direct effect OR a secondary response to the chronical loss of the protein, particularly when some mutants used in the work showed chronic growth defects. It would be useful to validate the main findings using degron alleles. For example, does degron-driven loss of Mec1/Tel1 also affected Rad54 foci levels less prominently than Rad53 loss?

3) Figure 2D and 2E, please provide quantification results of Rad54-GFP protein levels among the strains examined using biological duplicates.

4) Figure 4, did rtg2∆ rad53∆ and rtg3∆ rad53∆ cells show similar reduction of Rad54-GFP signals as rad53∆ cells? Could the authors speculate why rtg2∆ but not rtg3∆ worsened Rad54 signal reduction in mec1∆ cells? The elegant genetic data shown focused on negative phenotype; rescue or restoration of defects could provide strong support to the proposed model. For example, whether up-regulation of the Rtg pathway, such as through Rtg3 over-expression (which in principle could increase Rad53 activity), confers suppression of mec1∆sml1∆ cells?

5) Figure 5, did Rtg3 nuclear re-localization show dependence on Mec1, Tel1, or Rad53?

6) Figure 6, did rtg3∆ and 2∆ reduce Rad53 phosphorylation in Mec1+ Tel1+ cells or did they affect any HR proteins besides Rad54?

7) Discussion – Could the authors speculate whether kinases downstream of Rtg3 could directly phosphorylate Rad53 in mec1∆tel1∆ cells? Could there be possible connections between the Rad53-mediated histone regulation and the RTG pathway?

---

## [Author Response]

Essential revisions:1. Did rtg3∆ and 2∆ affect any other HR proteins besides Rad54? (mostly Reviewer 3 but also pointed out by Reviewer 1)

Interestingly, other HR proteins (such as Rad52-GFP) were in different clusters in our screen data. Furthermore, Rad52-GFP re-localization did not show much of a difference between *MEC1* and *RAD53* deletion mutants. Since our main findings in our manuscript focused on Rtg3 affecting Rad53- and not Mec1-dependent protein re-localizations (Rad54 being an example), we did not think Rtg3 would affect other HR proteins, since they were not in the same cluster as Rad54.

However, the question of how universal our findings are for Rtg3 is certainly of interest. Instead of focusing solely on HR proteins, we looked at seven proteins selected from the same cluster as Rad54. We identified five (Atg29, Dbf4, Mre11, Xbp1, and Msn1) proteins that had altered protein re-localization kinetics in *rtg3*∆ cells. The data are included in our manuscript as Figure 4F, we have added text to update our findings (lines 274-282), and a more detailed account of our experiment is described in our response to reviewer 3.

Added text:

“To complement our analysis Rad54 focus formation we asked whether the RTG pathway influences the re-localization kinetics of other proteins in the Rad53-dependent but Mec1-independent category (cluster *k*=2 in Figure 1C). We examined the re-localization of 7 proteins in cluster *k*=2 that had a detectable localization in the control condition and spanned distinct biological functions (Atg29, Dbf4, Exo1, Msn1, Mre11, Rpn4, and Xbp1) in *rtg3∆*. Five proteins (Atg29, Msn1, Mre11, Dbf4 and Xbp1) had reduced re-localization during MMS treatment when *RTG3* was deleted (Figure 4F). Thus, the RTG pathway promotes the Rad53-dependent localization changes of at least 5 additional proteins, suggesting that the RTG pathway activates Rad53 rather than acting directly on Rad54. We conclude that Rtg3 is a regulator of multiple proteins whose re-localization depends on Rad53, but not on Mec1.”

2. The readout used in the work was GFP signal intensity and not subcellular localization. The two could be correlated in some but not all cases. For example, increase in GFP intensity could reflect protein forming DNA repair foci or protein level increase. It will help increase the clarity of the work if the readout can be better defined in the text and the distinction can be made regarding changes of signal intensity vs protein localization (Reviewer 3).

We agree that GFP signal intensity and subcellular localization might not be correlated in all cases. We have added text to more clearly indicate how our computational approach minimizes the effects of changes in protein level (lines 116-118). Specifically, a key normalization is performed by dividing by the median cellular GFP intensity, to minimize the effect of protein abundance changes being identified as localization changes.

Added text:

“Importantly, the analysis method includes normalization to the median cellular GFP signal to minimize the influence of protein abundance changes on identification of localization change events.”

3. It can be difficult to discern whether the observed changes of GFP signal intensity in a mutant reflected a direct effect OR a secondary response to the chronical loss of the protein, particularly when some mutants used in the work showed chronic growth defects. It would be useful to validate the main findings using degron alleles. For example, does degron-driven loss of Mec1/Tel1 also affected Rad54 foci levels less prominently than Rad53 loss (Reviewer 3)?

We agree that the observed phenotypes could be chronic rather than acute, although we think the phenotypes are interesting in either case. Unfortunately, we were unable to complete the suggested experiments as when we constructed the *MEC1-AID* degron allele it was clearly a hypomorph. A detailed account of the experiments we performed, and the accompanying figures, are provided below in the responses to reviewer 3.

4. Figure 4, did rtg2∆ rad53∆ and rtg3∆ rad53∆ cells show similar reduction of Rad54-GFP signals as rad53∆ cells? Could the authors speculate why rtg2∆ but not rtg3∆ worsened Rad54 signal reduction in mec1∆ cells? (Reviewer 3)

We performed further experiments and found that *RTG2/RTG3* and *RAD53* double deletion mutants did not further reduce Rad54-GFP signals/foci in comparison to *rad53*∆ cells. These results have been added to Figure 4E, and the text has been updated to reflect these changes (lines 270-273).

We also speculate why *rtg2*∆ may show greater effects than *rtg3*∆, in our response to reviewer 3 below. Briefly, Rtg3 functions with Rtg1, both of which are regulated by Rtg2.

*rtg2*∆ cells may show greater reduction than *rtg3*∆ cells since Rtg2 is further upstream and not only impairs Rtg3 function, but also impairs Rtg1.

Added text:

“By contrast, *rtg2*Δ *rad53*Δ and *rtg3*Δ *rad53*Δ mutants did not have lower Rad54-GFP foci levels than *rad53*Δ (Figure 4E). Taken together, these results provide evidence that the *RTG* genes and *RAD53* function within the same pathway to regulate Rad54 focus formation.”

5. Rad53 phosphorylation have only been tested in mec1 background. It is important to show what is the contribution of Rtgs to Rad53 phosphorylation in a WT background (Reviewers 1 and 3).

We have provided additional data and included it as Figure 6 —figure supplement 1. Our results show that Rad53 phosphorylation, as assessed by gel mobility shift due to hyperphosphorylation and visualized by immunoblot, is not qualitatively different between the *RTG3* deletion mutant and control (WT). We updated the text to reflect these findings (lines 316-317).

Added text:

“Rad53 hyper-phosphorylation status did not appear to change in *rtg3*Δ cells, as assessed by mobility shift in immunoblot analysis (Figure 6 —figure supplement 1).”

6. Provide more timepoints for the FACS profiles to better highlight the effect of Rtg3 deletion in replication progression (Reviewer 1 and 2)

We repeated the FACS experiment for WT and rtg3∆ with additional time points to highlight the effect of the Rtg3 deletion on S phase progression, as well as to demonstrate that the results were reproducible (a concern for one reviewer). The repeated experiment is now Figure 5D, and the original panel is Figure 5 —figure supplement 1. The text has also been updated (lines 302-309).

Added text:

“Cells were arrested in G1 phase and released synchronously into S phase in the presence of MMS (Figure 5D, Figure 5 —figure supplement 1). As expected, *mec1*Δ and *rad53*Δ cells exhibited S phase defects, both progressing faster than wild type cells and reaching 2C DNA content by 60 minutes (Shimada et al., 2002), whereas wild type cells remain in S phase with less than 2C DNA contents. The *rtg3∆* cells progressed through S phase even more slowly than wild type. Although we cannot exclude some contribution of S phase entry delay, slow progression through S phase is consistent with the role of *RTG3* in replicating large replicons (Theis et al., 2010).”

7. Provide biological duplicates for Figure 2D and 2E (Reviewer 3)

We have performed replicates for the immunoblots presented in Figure 2D and 2E. The replicate for 2D is highly similar to the original and supports our inference that differences in Rad54 steady-state protein abundance do not contribute to the reduction in Rad54 foci seen in *rad53*∆ relative to *mec1*∆.

The replicate is included as Figure 2—figure supplement 1, and the quantification of the two blots, as requested by reviewer 3, is included as Figure 2E.

The replicate for 2E was quite noisy, as the level of Rad54 expressed from the SWE1pr is lower than the native level. We have not been able to get westerns of sufficient quality to allow quantification. As such, we have opted to remove the SWE1pr results, previously Figure 2E and 2F, from the manuscript. The text has been modified (lines 190-195) to reflect these changes.

Modified text:

“We found that Rad54 levels increase during MMS treatment, and that Rad54 abundance is lower in the checkpoint mutants than in the *sml1∆* control strain (Figure 2D, Figure 2 —figure supplement 1). However, Rad54 levels were similar in *mec1∆ tel1∆ sml1∆* and *rad53∆ sml1∆* cells (Figure 2E), indicating that the decreased Rad54 re-localization in *rad53∆ sml1∆* compared to *mec1∆ tel1∆ sml1∆* was not due to differences in Rad54 levels. We conclude that Rad54 focus formation is not strictly linked to Rad54 abundance.

The original (left) and replicate (right) are shown in Author response image 1, for reference.

**Author response image 1. sa2fig1:** 

Reviewer #1 (Recommendations for the authors):This study shows the power of yeast genetics when combined with modern quantitative imaging and mass spectrometry technology. I would ask the authors to address the following points prior to publication.1. Rad53 phosphorylation have only been tested in mec1 background. I think it would be an important information to the reader to show what is the contribution of Rtgs in an otherwise WT background. Even if under these conditions Mec1-dependent phosphorylation would dominate that would be important information.

We tested the contribution of Rtg3 to Rad53 phosphorylation in the WT background, by immunoblot analysis. As we, and the reviewer, suspected, the hyperphosphorylation and gel mobility shift of Rad53 was identical in WT (BY4741 background) and the *RTG3* deletion strain. This is likely due to Mec1 still being present, functional, and activating Rad53 in conditions of replication stress. The data is now added to our manuscript as Figure 6—figure supplement 1 and in the text, lines 316-317.

Added text:

“Rad53 hyper-phosphorylation status did not appear to change in *rtg3*Δ cells, as assessed by mobility shift in immunoblot analysis (Figure 6 —figure supplement 1).”

2. Despite the initial screening effort, the authors focus exclusively on Rad54 focus formation throughout the paper. Ideally, they would show some of the later findings of the paper for other factors to show universality of their findings. At least, however, they should discuss why they focused on Rad54 and what cellular activity they think Rad54 focus formation corresponds to.

The reviewer raises a good point in that our study focused exclusively on Rad54 focus formation, perhaps to the detriment of broader implications. We used Rad54 focus formation after MMS treatment largely as a reporter/read out for the Rad53-dependent and Mec1-independent pathway that we were interested in characterizing further. Rad54 focus formation was robust, easily distinguishable, and readily quantifiable, making it a great reporter for further assays. In addition, Rad54 has important roles in DNA repair, making it an attractive protein for more detailed analyses. We have expanded on our rationale for focusing on Rad54 (lines 173-177).

Added text:

“In subsequent assays, we use Rad54 re-localization as an exemplar of Mec1-independent regulation by Rad53. Rad54 foci form in MMS-induced replication stress and are easily distinguishable and readily quantifiable. Furthermore, we were interested in Rad54 since it has clear connections to DNA damage and repair, playing an important functional role in homologous recombination (HR), where multiprotein complexes assemble into nuclear foci (Lisby et al., 2004).”

Interestingly, our findings involving *RTG3* extend beyond Rad54 focus formation, and affect at least five other proteins of interest (i.e. Rad53-dependent and Mec1-independent). These results have been added as Figure 4F and provide evidence for the universality of our findings, with respect to Rtg3 protein function in replication stress response. The text has been updated to reflect these findings (lines 274-282).

Added text:

“To complement our analysis Rad54 focus formation we asked whether the RTG pathway influences the re-localization kinetics of other proteins in the Rad53-dependent but Mec1-independent category (cluster *k*=2 in Figure 1C). We examined the re-localization of 7 proteins in cluster *k*=2 that had a detectable localization in the control condition and spanned distinct biological functions (Atg29, Dbf4, Exo1, Msn1, Mre11, Rpn4, and Xbp1) in *rtg3∆*. Five proteins (Atg29, Msn1, Mre11, Dbf4 and Xbp1) had reduced re-localization during MMS treatment when *RTG3* was deleted (Figure 4F). Thus, the RTG pathway promotes the Rad53-dependent localization changes of at least 5 additional proteins, suggesting that the RTG pathway activates Rad53 rather than acting directly on Rad54. We conclude that Rtg3 is a regulator of multiple proteins whose re-localization depends on Rad53, but not on Mec1.”

3. Compared to the very high experimental standard throughout the paper, I was not too convinced by the assay measuring S phase progression and would suggest that it is repeated with more timepoints.

As noted in the ‘Essential revisions’ above, we repeated the FACS experiment for WT and *rtg3*∆ with additional time points to highlight the effect of the Rtg3 deletion on S phase progression, as well as to demonstrate that the results were reproducible (a concern for one reviewer). The repeated experiment is now Figure 5D, and the original panel is Figure 5 —figure supplement 1. The text has also been updated (lines 302-309).

Added text:

“Cells were arrested in G1 phase and released synchronously into S phase in the presence of MMS (Figure 5D, Figure 5 —figure supplement 1). As expected, *mec1*Δ and *rad53*Δ cells exhibited S phase defects, both progressing faster than wild type cells and reaching 2C DNA content by 60 minutes (Shimada et al., 2002), whereas wild type cells remain in S phase with less than 2C DNA contents. The *rtg3∆* cells progressed through S phase even more slowly than wild type. Although we cannot exclude some contribution of S phase entry delay, slow progression through S phase is consistent with the role of *RTG3* in replicating large replicons (Theis et al., 2010).”

Reviewer #2 (Recommendations for the authors):Figure 5C: The data for the control and RTG3-deletion background appear to be duplicated in the left and right panel of this figure. I recommend combining the two panels into a single panel to avoid such duplication. This should not diminish the clarity of the result presentation.

We agree that the duplicated control and *RTG3* mutant backgrounds were redundant. Figure 5C has been updated to combine these two panels into a single panel. The figure caption has also been updated to reflect these changes.

Figure 5D: The effect of RTG3 deletion on the FACS profiles is rather minor and may also be influenced by a delayed release from the α-factor arrest. How reproducible are these effects? Also: Since the FACS profiles of the DNA content are likely dominated by origin firing, have the authors tried to look for effects on origin activity after replication stress in these different genetic backgrounds? While not strictly required, this could provide further support for a role of Rtg3 in controlling S phase progression.

As noted in the ‘Essential revisions’ above, we repeated the FACS experiment for WT and rtg3∆ with additional time points to highlight the effect of the Rtg3 deletion on S phase progression, as well as to demonstrate that the results were reproducible (a concern for one reviewer). The repeated experiment is now Figure 5D, and the original panel is Figure 5 —figure supplement 1. The text has also been updated (lines 302-309).

Added text:

“Cells were arrested in G1 phase and released synchronously into S phase in the presence of MMS (Figure 5D, Figure 5 —figure supplement 1). As expected, *mec1*Δ and *rad53*Δ cells exhibited S phase defects, both progressing faster than wild type cells and reaching 2C DNA content by 60 minutes (Shimada et al., 2002), whereas wild type cells remain in S phase with less than 2C DNA contents. The *rtg3∆* cells progressed through S phase even more slowly than wild type. Although we cannot exclude some contribution of S phase entry delay, slow progression through S phase is consistent with the role of *RTG3* in replicating large replicons (Theis et al., 2010).”

We have not looked at origin firing in *RTG3* mutants after replication stress, compared to control, although we certainly agree that dissecting effects on origin firing vs fork rate would be very interesting.

Reviewer #3 (Recommendations for the authors):I have a few suggestions to help clarify methods used and data interpretation.1) The readout used in the work was GFP signal intensity and not subcellular localization. The two could be correlated in some but not all cases. For example, increase in GFP intensity could reflect protein forming DNA repair foci or protein level increase. It will help increase the clarity of the work if the readout can be better defined in the text and the distinction can be made regarding changes of signal intensity vs protein localization.

We agree that GFP signal intensity and subcellular localization might not be correlated in all cases. We have added text to more clearly indicate how our computational approach minimizes the effects of changes in protein level (lines 116-118). Specifically, a key normalization is performed by dividing by the median cellular GFP intensity, to minimize the effect of protein abundance changes being identified as localization changes.

Added text:

“Importantly, the analysis method includes normalization to the median cellular GFP signal to minimize the influence of protein abundance changes on identification of localization change events.”

2) It can be difficult to discern whether the observed changes of GFP signal intensity in a mutant reflected a direct effect OR a secondary response to the chronical loss of the protein, particularly when some mutants used in the work showed chronic growth defects. It would be useful to validate the main findings using degron alleles. For example, does degron-driven loss of Mec1/Tel1 also affected Rad54 foci levels less prominently than Rad53 loss?

The reviewer raises an interesting point. With the experiments that we have performed we cannot discern whether the changes and phenotypes that we see in our *mec1* mutants are an acute response or an adaptation. The Mec1-independent activation of Rad53 is seen with freshly constructed *mec1*∆, so we don’t think it is due to the longer-term passaging associated with the initial screen. The mediator mutants, which lack the growth defect seen in *mec1*∆, also show Rad53 activity. We have added extra text (lines 365-371) to our discussion to clarify that adaptation is also a possibility.

Added text:

“However, it remains to be seen whether Mec1-independent activation of Rad53 is present constitutively or is only active when canonical checkpoint signaling is compromised. At present we cannot distinguish between Mec1-independent activation of Rad53 being an acute response or a longer-term adaptation. We note, however, that the Mec1-independent Rad53 activation phenomenon is observed in independently constructed *mec1*∆ strains without a need for long-term passaging of cells and is observed in *rad9*∆ and *mrc1-AQ* contexts where cell growth is not compromised as it is in *mec1*∆.”

We attempted to distinguish between an acute response and adaptation using an AID degron approach, as suggested by the reviewer. Our approach was to construct a *mec1* degron allele in strains that also expressed Rad54-GFP, including *sml1∆* and *rad53∆ sml1∆*. Unfortunately, the *mec1-AID* allele in the context of the control (*sml1*∆ *mec1-AID*) was a hypomorph (MMS sensitive) without the addition of NAA to induce Mec1 degradation, as shown in Author response image 2. Considering that the *mec1-AID* strains were extremely sensitive to MMS before inducing Mec1 degradation, measurements in the mec1-AID background would not reflect an acute response.

3) Figure 2D and 2E, please provide quantification results of Rad54-GFP protein levels among the strains examined using biological duplicates.

As noted in the ‘Essential revisions’, we have performed replicates for the immunoblots presented in Figure 2D and 2E. The replicate for 2D is highly similar to the original and supports our inference that differences in Rad54 steady-state protein abundance do not contribute to the reduction in Rad54 foci seen in *rad53*∆ relative to *mec1*∆. The replicate is included as Figure 2—figure supplement 1, and the quantification of the two blots, as requested by reviewer 3, is included as Figure 2E.

The replicate for 2E was quite noisy, as the level of Rad54 expressed from the SWE1pr is lower than the native level. We have not been able to get westerns of sufficient quality to allow quantification. As such, we have opted to remove the SWE1pr results, previously Figure 2E and 2F, from the manuscript and from the accompanying text.

4) Figure 4, did rtg2∆ rad53∆ and rtg3∆ rad53∆ cells show similar reduction of Rad54-GFP signals as rad53∆ cells? Could the authors speculate why rtg2∆ but not rtg3∆ worsened Rad54 signal reduction in mec1∆ cells? The elegant genetic data shown focused on negative phenotype; rescue or restoration of defects could provide strong support to the proposed model. For example, whether up-regulation of the Rtg pathway, such as through Rtg3 over-expression (which in principle could increase Rad53 activity), confers suppression of mec1∆sml1∆ cells?

We thank the reviewer for raising an interesting question. One possible explanation for the strong defect in *rtg2*∆ compared to *rtg3*∆ is that Rtg3 forms a complex with another transcriptional activator, Rtg1. Both the nuclear localization of Rtg1 and Rtg3 are controlled by Rtg2. Therefore, it is possible that in the *rtg2*∆ context, both Rtg1 and Rtg3 are unable to exert their functions, leading to a more severe defect (in this case, decreased Rad54-GFP focus formation). Conversely, in the *rtg3*∆ context, Rtg1 function is unperturbed, supporting more Rad54 focus formation. A recent preprint also suggests that different phenotypes exist among *rtg1*∆, *rtg2*∆ and *rtg3*∆, in conditions of arginine deprivation and canavanine exposure (BioRxiv 2022, https://doi.org/10.1101/2022.02.21.481281).

We analyzed Rad54-GFP re-localization signal in *rtg2*∆*rad53*∆, *rtg3*∆*rad53*∆, and *rad53*∆ cells. In all instances, the localization of Rad54-GFP was reduced compared to wild-type, and *rtg2*∆ or *rtg3*∆ did not further reduce Rad54-GFP focus formation in *rad53*∆ mutant cells. This data has been added to Figure 4E. The text has been updated to reflect these changes (lines 270-271).

Added text:

“By contrast, *rtg2*Δ *rad53*Δ and *rtg3*Δ *rad53*Δ mutants did not have lower Rad54-GFP foci levels than *rad53*Δ (Figure 4E).

5) Figure 5, did Rtg3 nuclear re-localization show dependence on Mec1, Tel1, or Rad53?

While the reviewer poses an interesting question, Rtg3-GFP re-localization in replication stress was not assessed in the *mec1, tel1,* or *rad53* checkpoint mutant strains. Rtg3-GFP nuclear signal is rather weak in 0.03% MMS and so we did not pursue the genetic dependencies of Rtg3 localization further.”

6) Figure 6, did rtg3∆ and 2∆ reduce Rad53 phosphorylation in Mec1+ Tel1+ cells or did they affect any HR proteins besides Rad54?

As noted in the ‘Essential Revisions’, we tested the phosphorylation of Rad53 by western blot in *rtg3* deletion mutants in *MEC1 TEL1* wild-type background. The phosphorylation appears to be the same as in WT cells. This is likely because Rad53 phosphorylation by Mec1 predominates, causing a maximal mobility shift. We did not test *rtg3∆* compared to the control by mass spectrometry. It is possible that there are specific phosphorylation sites that are affected that have not been identified in our study.

These results have been added to the manuscript as Figure 6—figure supplement 1 and Figure 6-Source Data and the text has been modified on lines 316-317.

Added text:

“Rad53 hyper-phosphorylation status did not appear to change in *rtg3*Δ cells, as assessed by mobility shift in immunoblot analysis (Figure 6 —figure supplement 1).”

As noted in the ‘Essential Revisions’, we tested the effect of *rtg3∆* on the localization of other proteins that were identified as Rad53-dependent, similar to Rad54. We generated *RTG3* deletion mutants in 7 other GFP-tagged proteins. Five of these proteins (Atg29, Dbf4, Mre11, Xbp1, and Msn1) have a reduction in their replication stress-induced protein re-localization in an *RTG3* mutant background. We believe that this provides additional evidence for the replication stress related functions of *RTG3,* particularly in Rad53 activation, since the effects of Rtg3 appear to be more universal than we previously proposed. The data are included in our manuscript as Figure 4F and we have added text to update our findings (lines 274-282).

Added text:

“To complement our analysis Rad54 focus formation we asked whether the RTG pathway influences the re-localization kinetics of other proteins in the Rad53-dependent but Mec1-independent category (cluster *k*=2 in Figure 1C). We examined the re-localization of 7 proteins in cluster *k*=2 that had a detectable localization in the control condition and spanned distinct biological functions (Atg29, Dbf4, Exo1, Msn1, Mre11, Rpn4, and Xbp1) in *rtg3∆*. Five proteins (Atg29, Msn1, Mre11, Dbf4 and Xbp1) had reduced re-localization during MMS treatment when *RTG3* was deleted (Figure 4F). Thus, the RTG pathway promotes the Rad53-dependent localization changes of at least 5 additional proteins, suggesting that the RTG pathway activates Rad53 rather than acting directly on Rad54. We conclude that Rtg3 is a regulator of multiple proteins whose re-localization depends on Rad53, but not on Mec1.”

7) Discussion – Could the authors speculate whether kinases downstream of Rtg3 could directly phosphorylate Rad53 in mec1∆tel1∆ cells? Could there be possible connections between the Rad53-mediated histone regulation and the RTG pathway?

The kinases downstream of Rtg3 have certainly been of interest to us and would be of great interest to the field. We have made some efforts to identify a kinase that Rtg3 regulates that is involved in the Mec1-independent pathway of replication stress response, data that was not included in the manuscript.

Using the YEASTRACT+ database (http://www.yeastract.com/) we compiled 1189 potential protein targets of the Rtg3 transcription factor, based on over 1580 bibliographic references and 310 DNA binding sites (Teixeira, M.C. et al. 2023, doi: 10.1093/nar/gkac1041). Among these proteins, 37 were on our array of kinase-associated genes. Finally, among these 37 genes, 6 were identified in our screen as causing a reduction in Rad54-GFP focus formation in a *mec1∆ sml1∆* mutant background, 4 of which encode kinases (*CDC28*, *HOG1*, *STT4*, *PHO80*).

Stt4 does not have any known connections to histone regulation or the replication stress response. Stt4 is a phosphatidylinositol 4-kinase that regulates phosphatidylinositol 4-phosphate synthesis (Audhya, A., and Emr, S.D., 2002, doi: 10.1016/s1534-5807(02)00168-5; Baird, D. et al., 2008, doi: 10.1083/jcb.200804003) and has no known protein substrates.

Hog1 kinase is reported to control Rtg1/Rtg3 transcription complex activity, at least in conditions of osmotic stress. Importantly, Hog1 is required for proper Rtg3 localization to the nucleus (Ruiz-Roig, C., et al., 2012, doi: 10.1091/mbc.E12-04-0289). Therefore, decreases in Rad54-GFP focus formation in Hog1 mutants may be due to reduced Rtg3 function, rather than the direct phosphorylation of Rad53 by Hog1 itself.

Finally, Pho80 and Cdc28 are both cyclin-dependent kinases with connections to cell-cycle progression. In our initial validation studies of our hits from the kinome-associated screen, both Cdc28 and Pho80 mutants did not reduce Rad54-GFP foci to the levels of *RTG3* deletion mutants. Therefore, it unlikely that these kinases are individually regulated by Rtg3, at least in the context of Rad54 focus formation during MMS-induced replication stress.

We decided not to speculate on these kinases or any potential connections with histone regulation in our manuscript, focussing instead on studies that suggest RTG signalling roles in nucleotide metabolism and replication.